# Normalization-equivariant Diffusion Models: Learning Posterior Samplers From Noisy And Partial Measurements

## Abstract

Diffusion models (DMs) have rapidly emerged as a powerful framework for image generation and restoration, achieving remarkable perceptual quality. However, existing DMs are primarily trained in a supervised manner by using a large corpus of clean images. This reliance on clean data poses fundamental challenges in many real-world scenarios, where acquiring noise-free data is hard or infeasible, and only noisy and potentially incomplete measurements are available. While some methods are capable of training DMs using noisy data, they are generally effective only when the amount of noise is very mild or when some additional noise-free data is available. In addition, existing methods for training DMs from incomplete measurements require access to multiple complementary acquisition processes, an assumption that poses a significant practical limitation. Here we introduce the first approach for learning DMs for image restoration using only noisy measurement data from a single operator. As a first key contribution, we show that DMs, and more broadly minimum mean squared error denoisers, exhibit a weak form of scale equivariance linking rescaling in signal amplitude to changes in noise intensity. We then leverage this theoretical insight to develop a denoising score-matching strategy that generalizes robustly to noise levels lower than those present in the training data, thereby enabling the learning of DMs from noisy measurements. To further address the challenges of incomplete and noisy data, we integrate our method with equivariant imaging, a complementary self-supervised learning framework that exploits the inherent invariants of imaging problems, in order to train DMs for image restoration from single-operator measurements that are both incomplete and noisy. We validate the effectiveness of our approach through extensive experiments on image denoising, demosaicing, and inpainting, along with comparisons with the state of the art.

## 1 Introduction

Nearly all image data used in computer vision tasks come from a physical imaging system (e.g., digital camera, PET/CT scanner, etc.). These physical imaging systems span many disparate fields, from astronomy (Vojtekova et al., 2020) to medicine (Heckel et al., 2024). In each application, the crucial similarity is that the measurements from the imaging system are not always directly useful for downstream tasks. This can be due to the measurement process (e.g., mosaicing, blurring, Fourier encoding, etc.) and/or measurement noise.

Recovering the desired image $\mathbf{x} \in \mathbb{R}^n$ from (noisy) measurements $\mathbf{y} \in \mathbb{R}^m$ requires solving an inverse problem. This inversion process is typically ill-posed and some regularization is needed to obtain a robust inversion. Previously, handcrafted priors such as sparsity (Donoho, 2006; Lustig et al., 2007) were popular in many inversion schemes. With the growth of deep learning methods, many end-to-end networks for image recovery have been proposed (Hammernik et al., 2018; Agarwal et al., 2019; Liang et al., 2021). Recently, there has been interest in using deep generative models (Bora et al., 2017; Kawar et al., 2022; Jalal et al., 2021; Chung et al., 2024; Holden et al., 2022; González et al., 2022; Spagnoletti et al., 2025) to compute estimators or sample from the posterior distribution $p(\mathbf{x}|\mathbf{y})$. In both settings, a common framework is supervised learning, where a large corpus of clean data $\{\mathbf{x}_i\}_{i=1}^N \sim p(\mathbf{x})$ is available for training.

In practice, however, we often do not have direct access to samples from the distribution $p(\mathbf{x})$ but rather to measurement samples $\{\mathbf{y}_i\}_{i=1}^N \sim p(\mathbf{y})$ where $\mathbf{y}_i$ are generated from the measurement process of the imaging system of interest. Prior works have proposed methods for learning a variety of estimators using only measurement data. These works range from end-to-end methods (Yaman et al., 2020; Moran et al., 2019; Lehtinen et al., 2018; Tachella et al., 2025a; Monroy et al., 2025) to generative techniques (Bora et al., 2018; Daras et al., 2023; 2024b;a; Lu et al., 2025). A key similarity between many of these techniques, however, is an assumption of either access to some noise-free data and/or measurements arising from multiple imaging operators. These assumptions may not be met in practice, where all data are noisy and observed via a single measurement operator.

Herein, we propose a method for learning generative diffusion models capable of restoring corrupted images using only degraded measurements, obtained from multiple images processed through a common measurement operator. Our first key contribution demonstrates that diffusion models (DMs), and more broadly minimum mean squared error (MMSE) denoisers, exhibit a weak form of scale equivariance, wherein rescaling the signal amplitude induces a corresponding change in the noise level. Leveraging this theoretical insight, we introduce a denoising score-matching framework that enables the learning of denoisers at noise levels below those present in the measurements. These denoisers can then be integrated into a DM sampler, yielding reconstructions with higher perceptual quality compared to existing self-supervised methods. Furthermore, we extend our approach beyond denoising by incorporating equivariance to geometric transformations (e.g., translations and rotations), allowing us to learn DMs in the challenging setting where data are observed via a single incomplete measurement operator.

## 2 RELATED WORK

**Diffusion Modeling** The primary goal of generative modeling techniques is to learn a probability distribution from observed samples, i.e, $\{\mathbf{x}_i\}_{i=1}^N \sim p(\mathbf{x})$. This can be accomplished in a variety of ways (Kingma & Welling, 2022; Goodfellow et al., 2014; Ho et al., 2020). After learning this distribution, the model can be queried to generate new samples from $p(\mathbf{x})$ via a sampling procedure. The most popular methods currently are diffusion based approaches (Ho et al., 2020; Song et al., 2021; Karras et al., 2022). These techniques accomplish the task by training a deep neural network with parameters $\boldsymbol{\theta}$ on the supervised loss,

$$\min_{\boldsymbol{\theta}} \sum_{i=1}^N \mathcal{L}_{\text{SUP}}(\mathbf{x}, \boldsymbol{\theta}) \quad \text{where} \quad \mathcal{L}_{\text{SUP}}(\mathbf{x}, \boldsymbol{\theta}) = \mathbb{E}_{\boldsymbol{\eta}, \sigma_t} \| D_{\boldsymbol{\theta}}(\mathbf{x} + \sigma_t \boldsymbol{\eta}, \sigma_t) - \mathbf{x} \|^2, \tag{1}$$

with $\boldsymbol{\eta} \sim \mathcal{N}(\mathbf{0}, \mathbf{I})$ and $\sigma_t \sim p(\sigma)$, such that the learned network approximates the MMSE estimator $D_{\boldsymbol{\theta}}(\mathbf{x} + \sigma_t \boldsymbol{\eta}, \sigma_t) \approx \mathbb{E}\{\mathbf{x}|\mathbf{x} + \sigma_t \boldsymbol{\eta}\}$ at each noise level $\sigma_t > 0$. The probability distribution can be sampled by solving the following stochastic differential equation (SDE) in reverse time,

$$d\mathbf{x} = -2\dot{\sigma}_t \frac{D_{\boldsymbol{\theta}}(\mathbf{x}, \sigma_t) - \mathbf{x}}{\sigma_t} dt + \sqrt{2\dot{\sigma}_t \sigma_t} d\boldsymbol{\omega}_t, \tag{2}$$

where $\boldsymbol{\omega}_t$ is a Brownian noise process and $t \in (0, 1)$, using a variety of solvers (Karras et al., 2022).

**Self-Supervised Learning for Denoising** When the data needed to train restoration and diffusion models come from a measurement device, there is no true notion of a clean, ground-truth training set (Belthangady & Royer, 2019; Lehtinen et al., 2018). The two primary challenges in learning models solely from corrupted measurements are noise and rank-deficient measurement operators. In the simplest denoising problem, the data are corrupted by additive Gaussian noise, i.e.,

$$\mathbf{y} = \mathbf{x} + \sigma_n \boldsymbol{\eta}, \quad \boldsymbol{\eta} \sim \mathcal{N}(\mathbf{0}, \mathbf{I}), \tag{3}$$

where $\sigma_n$ is called the *measurement* noise level. Self-supervised methods for learning denoisers from a dataset of noisy measurements alone, $\{\mathbf{y}_i\}_{i=1}^N$, have been developed and used with great success. For example, Stein's unbiased risk estimator (SURE) (Stein, 1981) provides an approach to learn an unbiased estimate of the MMSE estimator with access only to noise-corrupted data (Metzler et al., 2020; Soltanayev & Chun, 2021; Aali et al., 2023) by minimizing an equivalent objective which only relies on the measurements (where div is the divergence operator):

$$\mathcal{L}_{\text{SURE}}(\mathbf{y}, \boldsymbol{\theta}) = \| \mathbf{y} - D_{\boldsymbol{\theta}}(\mathbf{y}, \sigma_n) \|^2 + 2\sigma_n^2 \operatorname{div} D_{\boldsymbol{\theta}}(\mathbf{y}, \sigma_n). \tag{4}$$

A key limitation of SURE is that it can only be used to obtain an unbiased estimate of the supervised denoising loss in (1) for noise levels at and above the measurement noise level (i.e., $\sigma_t \geq \sigma_n$). To learn diffusion processes with noisy data $\mathbf{y}$, however, we must be able to learn MMSE estimators below the measurement level $\sigma_t \leq \sigma_n$ as well. Recently, methods have been proposed to learn denoisers below the measurement noise level of the data (Daras et al., 2024b;a) by running the sampler during training and enforcing consistency, but they struggle when there are no clean data available to assist in training (Lu et al., 2025). In this paper, we show that *normalization-equivariance* can help bridge the gap to additionally learn below the measurement noise level without any clean data.

**Self-Supervised Learning for Linear Inverse Problems**    In the general linear inverse problem setting where data are corrupted by additive Gaussian noise, the measurements are given by

$$\mathbf{y} = \mathbf{A}\mathbf{x} + \sigma_n^2 \boldsymbol{\eta}, \quad \boldsymbol{\eta} \sim \mathcal{N}(\mathbf{0}, \mathbf{I}), \tag{5}$$

where $\mathbf{A} \in \mathbb{R}^{m \times n}$ is typically rank-deficient or highly ill-conditioned, and thus cannot be easily inverted. To overcome the rank-deficient operator, several end-to-end techniques have been proposed to train recovery methods from only corrupted measurement data (Yaman et al., 2020; Tachella et al., 2022), including generative models (Bora et al., 2018; Daras et al., 2023; Kawar et al., 2024; Kelkar et al., 2023; Park et al., 2025; Aali et al., 2025). Many of these techniques split the measurements and predict one subset of the measurements from a different (potentially disjoint) subset of measurements from the same sample. These measurement splitting techniques require access to measurement datasets $\{\mathbf{y}_i, \mathbf{A}_i\}_{i=1}^N$ with many different operators $\mathbf{A}_i \in \mathcal{A} \triangleq \{\mathbf{A}_1, \ldots, \mathbf{A}_G\}$. Intuitively, identifying a unique distribution $p(\mathbf{x})$ from measurements should be easier as $|\mathcal{A}|$ grows.

If, however, $|\mathcal{A}| = 1$ (i.e., there is only one measurement device) we must use additional information about the signal distribution $p(\mathbf{x})$ to assist in its recovery. The relatively mild assumption that the signal set of plausible images is invariant to a group of transformations $\{\mathbf{T}_g\}_{g=1}^G$, such as translations, flips and/or rotations, is often enough for learning from a single forward operator (Chen et al., 2022), as it allows us to create virtual operators in the following way:

$$\mathbf{y}_i = \mathbf{A}\mathbf{x} = \mathbf{A}\mathbf{T}_g^{-1}\mathbf{T}_g\mathbf{x} = \mathbf{A}_g\mathbf{x}'. \tag{6}$$

where $\mathbf{A}_g = \mathbf{A}\mathbf{T}_g$ and $\mathbf{x}' = \mathbf{T}_g\mathbf{x}$. In other words, we get $G$ virtual operators $\mathcal{A} = \{\mathbf{A}\mathbf{T}_1, \ldots, \mathbf{A}\mathbf{T}_G\}$ (Tachella et al., 2023). This invariance information can be enforced in a self-supervised way using the equivariant imaging loss (Chen et al., 2022).

**Normalization Equivariant Denoisers**    Previous lines of work have explored architecture changes for better noise level generalization in supervised denoisers (Mohan et al., 2020; Herbreteau et al., 2024). Recently, normalization-equivariant network architectures for image denoising were proposed in the supervised setting (Herbreteau et al., 2024). This work poses that certain conventional neural network components, such as ReLUs, should be removed and replaced with alternatives that retain normalization-equivariant properties over the network. This has been shown to lead to better generalization across noise levels in supervised training scenarios.

## 3    NORMALIZATION INVARIANT DENOISING

In this section, we focus on the Gaussian denoising problem. We present a new self-supervised loss for learning denoisers below the measurement noise level, $\sigma_n$, using noisy data alone, $\{\mathbf{y}_i = \mathbf{x}_i + \sigma_n\boldsymbol{\eta}_i\}_{i=1}^N$. The resulting denoisers can subsequently be used to implement diffusion samplers.

As stated above, the MMSE denoiser can be learned at the measurement noise level without access to noise-free data by using SURE (4). However, SURE does not yield reliable estimates for noise levels below that of the measurements; that is, it is only valid for $\sigma \geq \sigma_n$. To overcome this limitation, we assume that the denoiser satisfies the following normalization equivariance property:

$$\forall \alpha \in \mathbb{R}_+, \forall \mu \in \mathbb{R}, \ \mathrm{D}_{\boldsymbol{\theta}}(\alpha\mathbf{y} + \mu\mathbf{1}, \sigma\alpha) = \alpha\,\mathrm{D}_{\boldsymbol{\theta}}(\mathbf{y}, \sigma) + \mu\mathbf{1}, \tag{7}$$

for all $\mathbf{y}$ in the measurement distribution. If (7) is satisfied, then denoising at any noise level $\sigma'$ can be straightforwardly achieved by rescaling a single denoiser $\mathrm{D}_{\boldsymbol{\theta}}(\cdot, \sigma)$ trained at level $\sigma$, i.e., $\mathrm{D}_{\boldsymbol{\theta}}(\mathbf{y}, \sigma') = \frac{\sigma'}{\sigma}\mathrm{D}_{\boldsymbol{\theta}}(\frac{\sigma}{\sigma'}\mathbf{y}, \sigma)$. Equivariance can be enforced through architectural constraints as done in previous work (Herbreteau et al., 2024) with supervised denoising. Instead, we propose to

embed the normalization equivariance property into the original SURE loss by using the modified loss

$$\mathcal{L}_{\text{NE-SURE}}(\mathbf{y}, \boldsymbol{\theta}) = \mathbb{E}_{\alpha,\mu} \left\{ \|\alpha\mathbf{y} + \mu\mathbf{1} - D_{\boldsymbol{\theta}}(\alpha\mathbf{y} + \mu\mathbf{1}, \alpha\sigma_n)\|^2 + 2(\alpha\sigma_n)^2 \operatorname{div} D_{\boldsymbol{\theta}}(\alpha\mathbf{y} + \mu\mathbf{1}, \alpha\sigma_n) \right\}, \tag{8}$$

where $\alpha \sim \mathcal{U}(0,1)$ and $\mu \sim \mathcal{U}(0,1)$, as we find that this leads to better performance for self-supervised learning, and where incorporating $\mu$ improves generalization. In practice, we evaluate the expectation by sampling a random pair $(\alpha, \mu)$, and use Monte Carlo SURE (Ramani et al., 2008) to approximate the divergence, which requires an additional network evaluation. Other approximations can be similarly employed under our equivariance assumption (Monroy et al., 2025).

**Posterior Sampling**  After training a self-supervised, or supervised, denoiser we can sample from the posterior of noised measurement by initializing a reverse-time diffusion SDE processes at the measurement noise level $\sigma_n$ and running the SDE solver on (2) to some $\sigma_{\min}$.

**Understanding Normalization Invariance**  A valid question to ask is: Are there realistic priors whose associated MMSE denoisers verify this assumption? An initial answer to this question can be investigated by looking at the definition of a scale equivariant MMSE estimator:

$$D(\alpha\mathbf{y}, \alpha\sigma_n) = \mathbb{E}(\mathbf{x}|\alpha\mathbf{y}, \alpha\sigma_n) = \frac{\int_{\mathbb{R}^n} p(\mathbf{x})p(\alpha\mathbf{y}|\mathbf{x}, \alpha\sigma_n)\mathbf{x}d\mathbf{x}}{\int_{\mathbb{R}^n} p(\tilde{\mathbf{x}})p(\alpha\mathbf{y}|\tilde{\mathbf{x}}, \alpha\sigma_n)d\tilde{\mathbf{x}}} \tag{9}$$

$$= \frac{\int_{\mathbb{R}^n} p(\mathbf{x})p(\mathbf{y}|\frac{\mathbf{x}}{\alpha}, \sigma_n)\mathbf{x}d\mathbf{x}}{\int_{\mathbb{R}^n} p(\tilde{\mathbf{x}})p(\mathbf{y}|\frac{\tilde{\mathbf{x}}}{\alpha}, \sigma_n)d\tilde{\mathbf{x}}} \tag{10}$$

$$= \frac{1}{\alpha} \frac{\int_{\mathbb{R}^n} p(\alpha\mathbf{x})p(\mathbf{y}|\mathbf{x}, \sigma_n)\mathbf{x}d\mathbf{x}}{\int_{\mathbb{R}^n} p(\alpha\tilde{\mathbf{x}})p(\mathbf{y}|\tilde{\mathbf{x}}, \sigma_n)d\tilde{\mathbf{x}}} \tag{11}$$

where the first line uses that $p(\alpha\mathbf{y}|\mathbf{x}, \alpha\sigma_n) = p(\mathbf{y}|\frac{\mathbf{x}}{\alpha}, \sigma_n)$ for Gaussian noise, and the second line relies on a change of variables in both integrals. Inspecting (11), we see that choosing a positively homogeneous prior of order $k \in \mathbb{Z}$, i.e., if $p(\alpha\mathbf{x}) = \alpha^k p(\mathbf{x})$ for all $\mathbf{x} \in \mathbb{R}^n$ and all $\alpha > 0$, results in the normalization-equivariant condition in (7) for $\mu = 0$ (this result can be directly extended to the case $\mu \neq 0$ by additionally assuming that $p(\mathbf{x})$ is invariant under additive shifts).

Unfortunately, no proper prior can be positively homogeneous, since we would get $\int_{\mathbb{R}^n} p(\mathbf{x})d\mathbf{x} = \infty$. Nonetheless, as shown in the theorem below, there are well-defined priors whose MMSE estimators are close to being normalization-equivariant. Again, for clarity, we focus on the case $\mu = 0$; the extension to $\mu \neq 0$ follows directly.

**Theorem 3.1.** *Let* $D(\mathbf{y}, \sigma)$ *be the MMSE estimator to recover an unknown image* $\mathbf{x} \sim p(\boldsymbol{x})$ *from* $\mathbf{y} = \mathbf{x} + \sigma\boldsymbol{\eta}$ *with* $\boldsymbol{\eta} \sim \mathcal{N}(\mathbf{0}, \mathbf{I})$. *Assume that* $p(\mathbf{x})$ *admits a factorization* $p(\mathbf{x}) = p_1(\mathbf{x})p_2(\mathbf{x})$, *with* $p_1(\mathbf{x})$ *and* $p_2(\mathbf{x})$ *depending only on* $\|\mathbf{x}\|$ *and* $\mathbf{x}/\|\mathbf{x}\|$ *respectively, such that the normalized image* $\boldsymbol{x}/\|\mathbf{x}\|$ *is independent of* $\|\mathbf{x}\|$. *Also assume that* $U(\mathbf{x}) = -\log p(\mathbf{x})$ *is twice continuously differentiable. Then,* $\exists \sigma^\star > 0$ *such that for all* $\sigma, \sigma' \in (0, \sigma^\star)$, $D(\mathbf{y}, \sigma)$ *and* $D(\mathbf{y}, \sigma')$ *are approximately equivalent by rescaling, i.e.,*

$$\| D(\mathbf{y}, \sigma') - \tfrac{\sigma'}{\sigma} D(\tfrac{\sigma}{\sigma'}\mathbf{y}, \sigma)\| \leq \epsilon .$$

*where the error* $\epsilon > 0$ *depends on the degree of non-homogeneity of* $p_1$, *as measured by the Fisher divergence w.r.t. to* $p(\mathbf{x}|\mathbf{y}, \sigma)$ *between* $p_1$ *and the closest positively homogeneous function.*

*Proof.* Let $\mathcal{P}(\mathbb{R}^n)$ be the class of functions on $\mathbb{R}^n$ that are positively homogeneous and whose logarithm is twice continuously differentiable with Lipschitz continuous gradient. Assume that $p(\mathbf{x})$ admits a factorization $p(\mathbf{x}) = p_1(\mathbf{x})p_2(\mathbf{x})$, with $p_1(\mathbf{x})$ and $p_2(\mathbf{x})$ depending only on $\|\mathbf{x}\|$ and $\mathbf{x}/\|\mathbf{x}\|$ respectively, and let $\tilde{p}_1$ be the function in $\mathcal{P}(\mathbb{R}^n)$ that is closest to $p_1$ in the sense of the Fisher divergence w.r.t. the posterior $p(\mathbf{x}|\mathbf{y}, \sigma) \propto p(\mathbf{x})p(\mathbf{y}|\mathbf{x}, \sigma)$, i.e.,

$$\tilde{p}_1 = \underset{q \in \mathcal{P}(\mathbb{R}^n)}{\arg\min} \int_{\mathbb{R}^n} \|\nabla \log q(\mathbf{x}) - \nabla \log p_1(\mathbf{x})\|^2 p(\mathbf{x}|\mathbf{y}, \sigma)d\mathbf{x} .$$

Consider the approximation $\tilde{p}(\mathbf{x}) \propto \tilde{p}_1(\mathbf{x})p_2(\mathbf{x})$ of $p$, obtained by replacing the correct marginal $p_1$ by $\tilde{p}_1$, and denote by $\tilde{p}(\mathbf{x}|\mathbf{y}, \sigma) \propto \tilde{p}(\mathbf{x})p(\mathbf{y}|\mathbf{x}, \sigma)$ the associated posterior distribution. We view $\tilde{p}$ as

an operational prior that may be improper, but we assume that $\tilde{p}(\mathbf{x}|\mathbf{y}, \sigma)$ is well defined. Moreover, we denote by $\kappa_\sigma$ the Fisher divergence between the posteriors $p(\mathbf{x}|\mathbf{y}, \sigma)$ and $\tilde{p}(\mathbf{x}|\mathbf{y}, \sigma)$, given by

$$\kappa_\sigma = \int_{\mathbb{R}^d} \|\nabla \log p(\mathbf{x}|\mathbf{y}, \sigma) - \nabla \log \tilde{p}(\mathbf{x}|\mathbf{y}, \sigma)\|^2 \, \pi(\mathbf{x}|\mathbf{y}, \sigma) d\mathbf{x} \,,$$

$$= \int_{\mathbb{R}^d} \|\nabla \log p_1(\mathbf{x}) - \nabla \log \tilde{p}_1(\mathbf{x})\|^2 \, \pi(\mathbf{x}|\mathbf{y}, \sigma) d\mathbf{x} \,,$$

where we have used the factorization property of $p$ and $\tilde{p}$.

Furthermore, because $\mathbf{x} \mapsto \log p(\mathbf{y}|\mathbf{x}, \sigma)$ is $1/\sigma^2$-strongly concave, and the Hessian of $\log \tilde{p}$ is bounded, there exists some $\sigma^\star$ such that for all $\sigma \leq \sigma^\star$ the approximation $\log \tilde{p}(\boldsymbol{x}|\boldsymbol{y}, \sigma)$ is strongly concave outside some compact set (i.e., for some constants $K > 0$ and $R \geq 0$, $\nabla^2 \log \tilde{p}(\mathbf{x}|\mathbf{y}) \succeq K\mathbf{I}$ for all $\|\mathbf{x}\| \geq R$). From (Huggins et al., 2018, Theorem 5.3), this implies that for any $\sigma \leq \sigma^\star$, the 2-Wasserstein distance between $p(\mathbf{x}|\mathbf{y}, \sigma)$ and $\tilde{p}(\mathbf{x}|\mathbf{y}, \sigma)$ is bounded as

$$\mathcal{W}_2 \left( p(\mathbf{x}|\mathbf{y}, \sigma), \tilde{p}(\mathbf{x}|\mathbf{y}, \sigma) \right) \leq \psi \kappa_\sigma \,,$$

where $\psi > 0$ depends on $\tilde{p}(\mathbf{x}|\mathbf{y}, \sigma)$, but is independent of $p(\mathbf{x}|\mathbf{y}, \sigma)$. For example, in the specific case where $\tilde{p}(\mathbf{x}|\mathbf{y}, \sigma)$ is strongly log-concave, $\psi$ is the inverse of the strong log-concavity constant.

Following on from this, we denote by $\tilde{\mathrm{D}}(\mathbf{y}, \sigma)$ the MMSE denoiser associated with $\tilde{p}(\mathbf{x}|\mathbf{y}, \sigma)$ and use (11) to show that $\tilde{\mathrm{D}}(\mathbf{y}, \sigma)$ verifies the desired rescaling property $\tilde{\mathrm{D}}(\mathbf{y}, \sigma') = \frac{\sigma'}{\sigma} \tilde{\mathrm{D}}(\frac{\sigma}{\sigma'}\mathbf{y}, \sigma)$. Lastly, because $\mathcal{W}_2$ bounds the difference in the expectation of random variables, we have

$$\|\mathrm{D}(\mathbf{y}, \sigma') - \tilde{\mathrm{D}}(\mathbf{y}, \sigma')\| = \|\mathrm{D}(\mathbf{y}, \sigma') - \frac{\sigma'}{\sigma}\tilde{\mathrm{D}}(\frac{\sigma}{\sigma'}\mathbf{y}, \sigma)\| \leq \psi \kappa_{\sigma'} \,,$$

$$\|\frac{\sigma'}{\sigma}\mathrm{D}(\frac{\sigma}{\sigma'}\mathbf{y}, \sigma) - \frac{\sigma'}{\sigma}\tilde{\mathrm{D}}(\frac{\sigma}{\sigma'}\mathbf{y}, \sigma)\| \leq \frac{\sigma'}{\sigma}\psi \kappa_\sigma \,,$$

and therefore,

$$\|\mathrm{D}(\mathbf{y}, \sigma') - \frac{\sigma'}{\sigma}\mathrm{D}(\frac{\sigma}{\sigma'}\mathbf{y}, \sigma)\| \leq \epsilon \,,$$

with $\epsilon^2 = \psi^2 \kappa_{\sigma'}^2 + (\frac{\sigma'}{\sigma})^2 \psi^2 \kappa_\sigma^2$, concluding the proof. $\qquad\square$

To develop an intuition for Theorem 3.1, it is helpful to consider the following. For most images, the magnitude $\|\mathbf{x}\|$ conveys negligible information about the normalized image $\mathbf{x}/\|\mathbf{x}\|$. Indeed, recovering $\mathbf{x}$ from the total energy $\|\mathbf{x}\|^2$ alone is typically impossible. Therefore, the assumption that $p(\mathbf{x})$ admits the proposed factorization $p(\mathbf{x}) = p_1(\mathbf{x})p_2(\mathbf{x})$ is practically justified. In addition, it has been repeatedly empirically observed that the spectral properties of images exhibit power-law statistics related to self-similarity (see, e.g., (Ruderman, 1997) and Figure 13 in Appendix A). This motivates the consideration of power-law models of the form $p_1(\mathbf{x}) = q(\mathbf{x})L(\mathbf{x})$ where $q$ is positively homogeneous, e.g., $q(\mathbf{x}) = \|\mathbf{x}\|^{-\beta}$, and $L(\mathbf{x})$ is some slow-varying function that predominantly influences $p_1(\mathbf{x})$ near the origin. In this case, minimizing the considered Fisher divergence leads to $\tilde{p}_1 = q$, and the remaining error, $\kappa_{\sigma_n} = \int_{\mathbb{R}^n} L(\mathbf{x})p(\mathbf{x}|\mathbf{y}, \sigma)d\mathbf{x}$, is small if $L(\mathbf{x})$ is small in regions with high posterior mass. To conclude, because training by score-matching is equivalent to minimizing the Fisher divergence w.r.t. $p(\mathbf{x}|\mathbf{y}, \sigma)$, we view performing score-matching subject to the considered normalization property as a mechanism for learning the approximate MMSE denoiser $\tilde{\mathrm{D}}$ rather than $\mathrm{D}$, and we expect this denoiser trained directly from noisy data to generalize robustly to lower noise levels provided $\sigma_n \leq \sigma^\star$ in Theorem 3.1, as evidenced by the experiments reported in Section 4.

## 4 DENOISING EXPERIMENTS

**Baselines** We evaluate the performance of the normalization-equivariance strategy below the measurement noise level by comparing to **(1)** a denoiser trained in supervised fashion at all noise levels **(2)** a self-supervised denoiser trained using SURE only at the measurement noise level (SURE) to understand how seeing only the measurement noise level generalizes performance (Metzler et al., 2020), and **(3)** a recently proposed method for self-supervised denoising below the measurement noise level using consistency (CD) (Daras et al., 2024a).

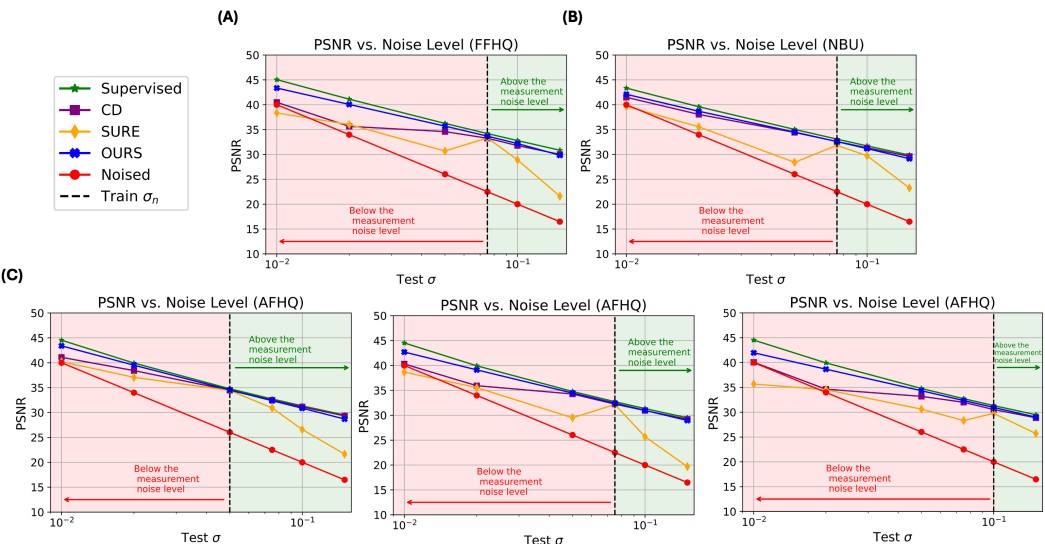

Figure 1: Performance of various one-step denoisers on (A) FFHQ (B) NBU and (C) AFHQ dataset.

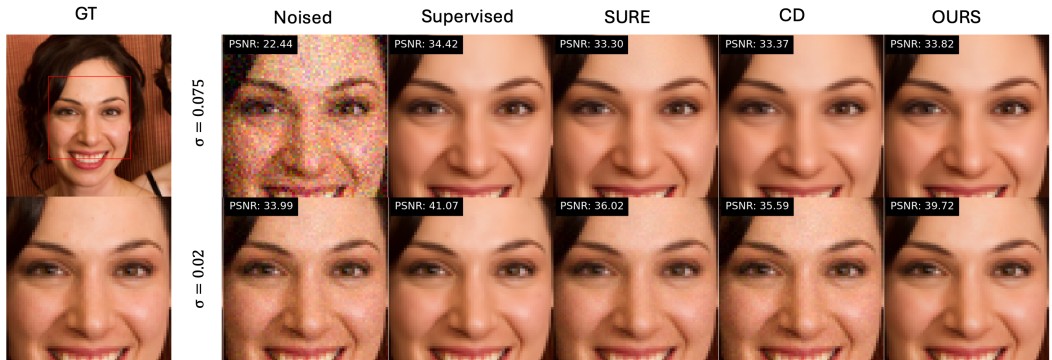

Figure 2: Denoising example on FFHQ dataset at test noise levels $\sigma = 0.075$ (top) and $\sigma = 0.02$ (bottom). All models, except for supervised, were trained on only noisy data with $\sigma_n = 0.075$.

**Training Details**   We compare the performance of all denoisers on the task of denoising FFHQ ($128 \times 128$ pixels) training data that has been corrupted with $\sigma_n = 0.075$. On the AFHQ ($128 \times 128$ pixels) dataset, we perform experiments for varying noise levels $\sigma_n \in \{0.05, 0.075, 0.10\}$. Finally, we use a panchromatic satellite imaging dataset (NBU) (Meng et al., 2020) ($128 \times 128$ pixels) corrupted with $\sigma_n = 0.075$. To investigate the performance of our technique with varying dataset sizes, we create various training set sizes of $N \in \{500, 1000, 5000, 15000\}$ for $\sigma_n = 0.05$ using the AFHQ dataset. We emphasize that for all datasets, we simulated noise only once for each image in the training dataset to most accurately emulate a real-world setting. Our model architecture is the UNet denoiser from Song et al. (2021) with 55M parameters.

**Sampler Details**   For experiments on diffusion sampling, after training each model, we run Algorithm 1 in Appendix A with $\mathbf{A} = \mathbf{I}$ from the measurement noise level to $\sigma_{\min} = 0.01$ (for experiments with $\sigma_n \in \{0.05, 0.075\}$) or $\sigma_{\min} = 0.02$ (for experiments with $\sigma_n = 0.10$). We use $K = 25$ steps for all sampling methods, and the timestep schedule following Karras et al. (2022):

$$\sigma_i = \left(\sigma_{\max}^{\frac{1}{\gamma}} + \frac{i}{N-1}\left(\sigma_{\min}^{\frac{1}{\gamma}} - \sigma_{\max}^{\frac{1}{\gamma}}\right)\right)^{\gamma}, \tag{12}$$

with $\gamma = 7$ and $\sigma_{\max} = \sigma_n$.

Table 1: Posterior sampling metrics on FFHQ, NBU, and AFHQ $128 \times 128$ where the models are trained and complete inference on data with additive noise shown in the 2nd column.

| Dataset | $\sigma_n$ | Solver | Sampler | Self Sup. | PSNR (↑) | SSIM (↑) | LPIPS (↓) | FID (↓) |
|---------|-----------|--------|---------|-----------|----------|----------|-----------|---------|
| FFHQ | 0.075 | MMSE (Self Sup.) | | ✓ | 33.68 | 0.937 | 0.024 | 29.82 |
| | | OURS | ✓ | ✓ | 32.76 | 0.921 | 0.015 | 22.57 |
| | | CD (Daras et al., 2024a) | ✓ | ✓ | 28.92 | 0.756 | 0.057 | 59.37 |
| | | MMSE (Sup.) (Karras et al., 2022) | | | **34.20** | **0.944** | 0.023 | 32.79 |
| | | EDM (Sup.) (Karras et al., 2022) | ✓ | | 33.06 | 0.920 | **0.012** | **19.91** |
| NBU | 0.075 | MMSE (Self Sup.) | | ✓ | 32.58 | 0.863 | 0.111 | 79.20 |
| | | OURS | ✓ | ✓ | 32.05 | 0.851 | 0.085 | 34.01 |
| | | CD (Daras et al., 2024a) | ✓ | ✓ | 32.02 | 0.836 | 0.076 | 53.75 |
| | | MMSE (Sup.) (Karras et al., 2022) | | | **33.08** | **0.877** | 0.110 | 103.18 |
| | | EDM (Sup.) (Karras et al., 2022) | ✓ | | 32.02 | 0.841 | **0.057** | **18.21** |
| AFHQ | 0.05 | MMSE (Self Sup.) | | ✓ | 34.52 | 0.949 | 0.020 | 10.09 |
| | | OURS | ✓ | ✓ | 33.89 | 0.942 | 0.011 | 5.08 |
| | | CD (Daras et al., 2024a) | ✓ | ✓ | 32.48 | 0.903 | 0.015 | 10.74 |
| | | MMSE (Sup.) (Karras et al., 2022) | | | **34.77** | **0.952** | 0.019 | 9.83 |
| | | EDM (Sup.) (Karras et al., 2022) | ✓ | | 34.01 | 0.942 | **0.010** | **3.62** |
| AFHQ | 0.075 | MMSE (Self Sup.) | | ✓ | 32.41 | 0.923 | 0.034 | 11.56 |
| | | OURS | ✓ | ✓ | 31.71 | 0.911 | 0.021 | 7.12 |
| | | CD (Daras et al., 2024a) | ✓ | ✓ | 29.44 | 0.818 | 0.034 | 17.08 |
| | | MMSE (Sup.) (Karras et al., 2022) | | | **32.72** | **0.928** | 0.034 | 12.55 |
| | | EDM (Sup.) (Karras et al., 2022) | ✓ | | 31.80 | 0.910 | **0.017** | **5.66** |
| AFHQ | 0.10 | MMSE (Self Sup.) | | ✓ | 30.96 | 0.898 | 0.053 | 13.80 |
| | | OURS | ✓ | ✓ | 30.38 | 0.888 | 0.034 | 8.88 |
| | | CD (Daras et al., 2024a) | ✓ | ✓ | 27.01 | 0.732 | 0.064 | 20.65 |
| | | MMSE (Sup.) (Karras et al., 2022) | | | **31.34** | **0.906** | 0.049 | 14.37 |
| | | EDM (Sup.) (Karras et al., 2022) | ✓ | | 30.26 | 0.879 | **0.024** | **7.82** |

We report all metrics over validation sets of 1000, 1500, 1800 images for FFHQ, AFHQ, and NBU datasets, respectively. To assess distortion in the reconstructed images, we report PSNR and SSIM (Wang et al., 2004), while for perceptual quality, we report LPIPS and FID.

**Denoising Performance** Figure 1 shows how each denoiser performs at noise levels different than the measurement noise level available at training time. Both SURE and CD perform poorly at noise levels below the measurement noise level. In contrast, our approach incurs only a small performance reduction with respect to the supervised case at noise levels below the training data noise, providing strong empirical support for the validity of the normalization invariance assumption. Moreover, for highly noisy training data, the gap between the proposed method and supervised learning at lower noise increases, but it remains the best method across self-supervised techniques. Figure 2 (top) show examples of how all self-supervised denoisers perform well at the measurement noise level compared to the supervised approach. However, we observe in Figure 2 (bottom) that, at noise levels below the measurement noise level, only our technique maintains a good performance compared to the supervised strategy.

**Posterior Sampling Performance** We assess each denoiser's sampling ability compared to a supervised denoiser in Table 1. Here, MMSE (Self Sup.) is taken to be the one-step denoiser from our normalization-equivariant SURE denoiser, as it showed better PSNR metrics than SURE alone in the one-step denoising task above (i.e., better approximate self-supervised MMSE estimator). The sampler based on our self-supervised denoiser reports better perceptual quality metrics than all methods except for the supervised denoiser. As expected, all sampling approaches report worse PSNR and SSIM than their respective MMSE (one-step) solvers, which is in line with the known perception-distortion tradeoff (Blau & Michaeli, 2018). In Figure 3 we show visual examples comparing each approach. Additionally, in Appendix A we show Figure 9 which are examples of each method at various training + inference noise levels on the AFHQ dataset along with each method's radial spectra for the corresponding reconstructions in Figure 10. All one-step solvers appear to reduce high frequency components resulting in a visually smoother image and CD retains noise in the sample leading to higher frequency components. Our method, however, produces spectra noticeably more closely aligned with the ground truth images.

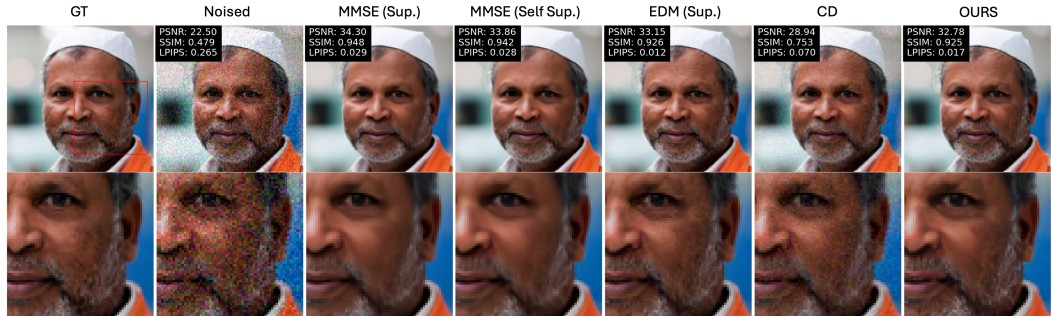

Figure 3: Example of various denoisers using diffusion sampling (except for MMSE (Self Sup.) and MMSE (Sup.) columns) on FFHQ dataset with training and test noise level $\sigma_n = 0.075$.

**Sample Complexity**   Figure 4 shows the performance of our denoiser trained on the AFHQ dataset as a function of the number of (noisy) samples available for training. We plot the difference between the average MSE loss for each denoiser and the MMSE performance (approximated here by supervised learning on the largest dataset size). Previous work (Daras et al., 2024a) hypothesized that learning below the measurement noise level requires a potentially prohibitive amount of noisy training data. We find, on the contrary, that the mean squared error scales approximately as $\sigma_t^2/N^{1/3}$ for noise levels above ($\sigma_t \geq 0.05$) and also below ($\sigma_t < 0.05$) the measurement noise, which is the complexity typically observed in supervised learning (Hardt & Recht, 2022, Ch. 6).

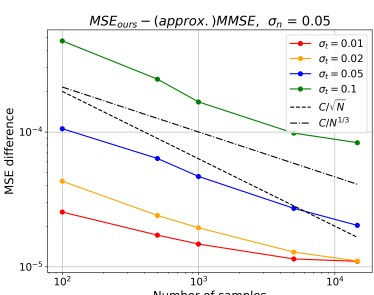

Figure 4: Denoiser performance on AFHQ with varying amounts of only noised training samples at $\sigma_n = 0.05$.

## 5   EXTENSION TO LINEAR INVERSE PROBLEMS

We now extend our method to more general linear inverse problems. In particular, we present a new self-supervised loss for learning DMs from a dataset of noisy measurements, all observed by the same forward operator at the same measurement noise level, $\{\mathbf{y}_i = \mathbf{A}\mathbf{x}_i + \sigma_n\boldsymbol{\eta}_i\}_{i=1}^N$, where $\mathbf{A}$ is rank deficient and $\boldsymbol{\eta} \sim \mathcal{N}(\mathbf{0}, \mathbf{I})$.

Following the equivariant imaging (EI) approach, we assume that $p(\mathbf{x})$ is invariant to a group of transformations $\{\mathbf{T}_g \in \mathbb{R}^{n \times n}\}_{g=1}^G$ such as translations, flips and/or rotations, that is, for any image $\mathbf{x} \sim p(\mathbf{x})$, $\mathbf{T}_g\mathbf{x}$ follows the same distribution. We propose the following loss to approximately learn $\mathrm{D}_{\boldsymbol{\theta}}(\mathbf{y}, \sigma) \approx \mathbb{E}\{\mathbf{x}|\mathbf{A}\mathbf{x} + \sigma\boldsymbol{\eta}\}$ in a self-supervised fashion for all noise levels $\sigma \in (0, \sigma_n]$:

$$\mathcal{L}(\mathbf{y}, \boldsymbol{\theta}) = \mathbb{E}_{\alpha,\mu} \left\{ \|\alpha\mathbf{y} + \mu\mathbf{1} - \mathrm{D}_{\boldsymbol{\theta}}(\alpha\mathbf{y} + \mu\mathbf{1}, \alpha\sigma_n)\|^2 + 2(\alpha\sigma_n)^2 \operatorname{div} \mathrm{D}_{\boldsymbol{\theta}}(\alpha\mathbf{y} + \mu\mathbf{1}, \alpha\sigma_n) \right\}$$

$$+ \mathbb{E}_{g,\sigma',\eta} \left\{ \|\mathbf{T}_g\hat{\mathbf{x}}_{\boldsymbol{\theta}} - \mathrm{D}_{\boldsymbol{\theta}}(\mathbf{A}\mathbf{T}_g\hat{\mathbf{x}}_{\boldsymbol{\theta}} + \sigma'\boldsymbol{\eta}, \sigma')\|^2 \right\} \tag{13}$$

where $\hat{\mathbf{x}}_{\boldsymbol{\theta}} = \mathrm{D}_{\boldsymbol{\theta}}(\mathbf{y}, \sigma_n)$, $\sigma' \sim \mathcal{U}(0, \sigma_n)$, $\boldsymbol{\eta} \sim \mathcal{N}(\mathbf{0}, \mathbf{I})$. The first two terms are the same as in the denoising setting (8) and allow generalizing to noise levels below the measurement level $\sigma_n$, and the last term is the EI loss (Chen et al., 2022) which allows learning in the nullspace of $\mathbf{A}$.

**Posterior Sampling**   The learned network can be used to approximate the MMSE denoiser in measurement space as $\mathbb{E}\{\mathbf{A}\mathbf{x}|\mathbf{A}\mathbf{x} + \sigma\boldsymbol{\eta}\} = \mathbf{A}\mathbb{E}\{\mathbf{x}|\mathbf{y}\} \approx \mathbf{A}\,\mathrm{D}_{\boldsymbol{\theta}}(\mathbf{y}, \sigma)$. Thus, as in the denoising setting in Section 4, we can run the reverse SDE in (2) in measurement space using the denoiser $\mathbf{A} \circ \mathrm{D}_{\boldsymbol{\theta}}$ from $\sigma_n$ to $\sigma_{\min}$, to sample an (almost) noiseless measurement $\mathbf{z} \sim p(\mathbf{A}\mathbf{x}|\mathbf{A}\mathbf{x} + \sigma_n\boldsymbol{\eta})$, and finally obtain an image space posterior sample as $\mathbf{x} = \mathrm{D}_{\boldsymbol{\theta}}(\mathbf{z}, \sigma_{\min})$. The resulting algorithm is summarized in Appendix A with Algorithm 1.

Table 2: Image inpainting (top) and Demosaicing (bottom) metrics on AFHQ $64 \times 64$.

| Task | $\sigma_n$ | Solver | Sampler | Self Sup. | PSNR (↑) | SSIM (↑) | LPIPS (↓) | FID (↓) |
|------|-----------|--------|---------|-----------|----------|----------|-----------|---------|
| Inpainting | 0.075 | EI (Chen et al., 2022) | | ✓ | **27.92** | **0.867** | 0.043 | 25.259 |
| | | OURS | ✓ | ✓ | 27.74 | 0.864 | **0.031** | **21.114** |
| | | Ambient Diff. (Daras et al., 2023) | ✓ | ✓ | 25.21 | 0.783 | 0.067 | 44.921 |
| Demosaic | 0.075 | EI (Chen et al., 2022) | | ✓ | **26.48** | **0.836** | 0.067 | 45.426 |
| | | OURS | ✓ | ✓ | 26.05 | 0.835 | **0.050** | **42.289** |
| | | Ambient Diff. (Daras et al., 2023) | ✓ | ✓ | 7.77 | 0.078 | 0.532 | 232.951 |

If the operator $\mathbf{A}$ is injective over the support of $p(\mathbf{x})$, that is, if reconstructions in the noiseless case are exact (see Tachella et al. (2022) for technical details), then the final reconstruction step yields a sample of the correct posterior distribution $p(\mathbf{x}|\mathbf{y})$.

## 5.1 EXPERIMENTS

**Baselines** We compare our approach to EI (Chen et al., 2022), a point estimator, with the same group of transformations. Additionally, we compare against Ambient Diffusion (Daras et al., 2023) trained without additive noise

**Training Details** We test the extension of our invariant sampler to non-trivial forward operators by solving inpainting and demosaicing tasks. For the inpainting task, we randomly simulated one inpainting mask with an undersampling factor of $m/n = 0.7$. After applying the same inpainting mask to all samples in the datasets, we added noise $\sigma_n = 0.075$ to the measurement data. Due to the random structure of the inpainting mask, we use the translational invariance of natural images and choose $\mathbf{T}_g$ to represent all circular shifts of the image. For demosaicing, we prepare our training datasets by selecting the Bayer filter and applied the corresponding $\mathbf{A}$ to each image in the training dataset and added noise $\sigma_n = 0.075$. We assumed rotational invariance of the underlying signal set and selected the group of transformations $\mathbf{T}_g$ that represents all possible rotations. For both inpainting and demosaicing experiments, we used a training set size of 15000 from AFHQ ($64 \times 64$ pixels). We utilize the DeepInverse (Tachella et al., 2025b) package for applying transformations in the equivariant imaging loss.

**Inference Details** For inference, we use Algorithm 1 with the same parameters as in Section 4, except we set $\mathbf{A}$ to the respective measurement operator.

**Sampler Performance** In Table 2, we show quantitative results comparing each self-supervised approach on inpainting and demosaicing. In both settings, EI retains superior distortion metrics (as expected for a point estimator analogous to an MMSE estimator (Blau & Michaeli, 2018)), but our approach beats both EI and Ambient Diffusion in perceptual quality metrics. Ambient Diffusion performs poorly because it lacks access to measurements from different measurement operators that cover the full ambient space. This means that at training time, there is a non-trivial nullspace that the model is never given self-supervised guidance to predict. We provide example restorations in the Appendix A(Figures 11 and 12).

## 6 CONCLUSION

We present a method for learning a denoiser from noisy data alone. In particular, the proposed method can learn below the noise level of the data, which is necessary for obtaining posterior samples with DMs. Our denoiser performs on par with supervised learning in noise levels below the measurement noise and can be used in diffusion sampling schemes. Moreover, our method can be extended to learn from noisy and incomplete measurements associated with a single degradation operator, producing higher perceptual quality images than existing self-supervised generative techniques. Future work will include extending this method to data with unknown noise levels (Tachella et al., 2025a), other measurement processes (e.g., MRI), as well as leveraging model architectures specific to image restoration (e.g., unrolled networks).

## REPRODUCIBILITY STATEMENT

To allow complete reproducibility, we commit to publishing the full code upon acceptance.

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
