# A APPENDIX

## A.1 ONE-STEP DENOISING EXPERIMENTS

We have included additional figures for our one-step denoiser experiments. Figures 5 and 6 show the one step denoiser performance of the various methods at the measurement noise level and a lower test noise level for the AFHQ and NBU datasets respectively. We observe that our method performs better at denoising below the measurement noise level compared to other self-supervised denoising techniques.

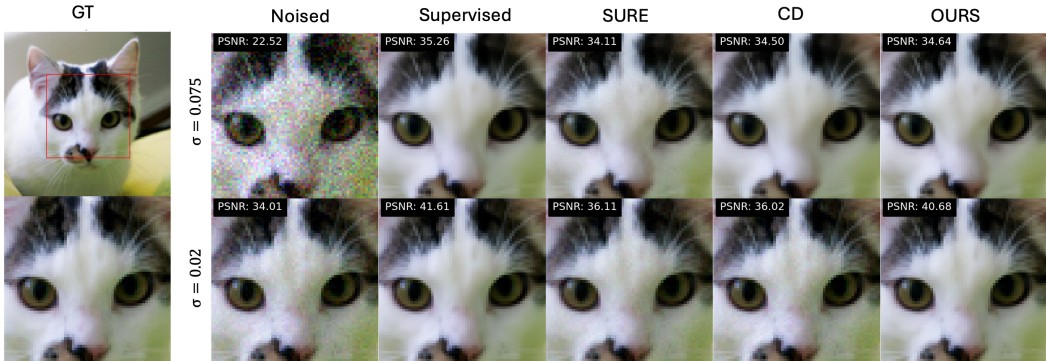

Figure 5: Example restorations of various denoisers on AFHQ dataset at test noise levels $\sigma_t = 0.075$ (top) and $\sigma_t = 0.02$ (bottom). All models, except for supervised were trained on only noisy data with $\sigma_n = 0.075$.

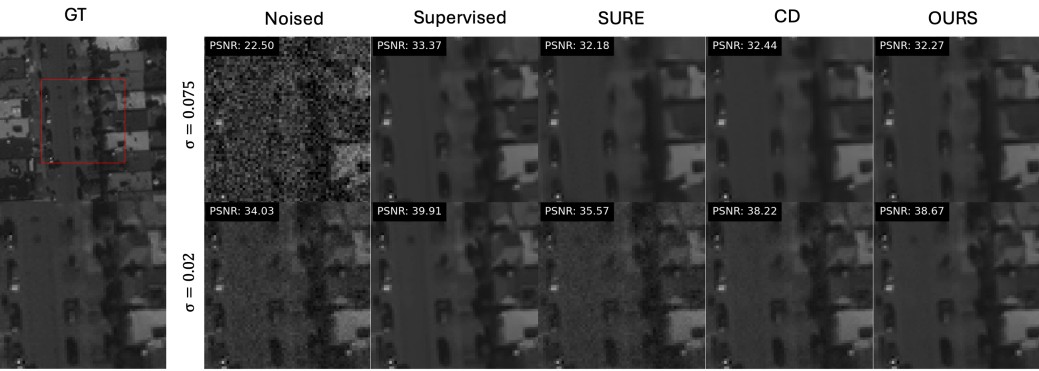

Figure 6: Example restorations of various denoisers on NBU dataset at test noise levels $\sigma_t = 0.075$ (top) and $\sigma_t = 0.02$ (bottom). All models, except for supervised were trained on only noisy data with $\sigma_n = 0.075$.

## A.2 DIFFUSION SAMPLING

Additional figures have been provided for diffusion sampling experiments on the AFHQ and NBU datasets. Figures 7 and 8 show example diffusion samples for supervised and self-supervised approaches discussed in the paper on AFHQ and NBU respectively. Figure 9 shows diffusion samples for different training + inference noise levels with accompanying radial spectrum plots in Figure 10. Here we see that while one-step supervised and self-supervised MMSE denoisers tend to reduce high frequency features, our method retains higher frequencies lending to our method providing better perceptual images.

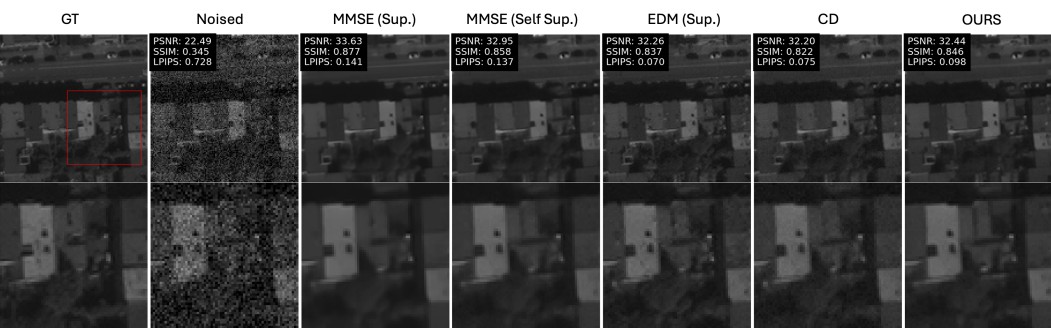

Figure 7: Example of various denoisers using diffusion sampling (except MMSE(Self Sup.) and MMSE(Sup.) columns) on AFHQ dataset with training and test noise level $\sigma_n = \sigma_t = 0.075$.

Figure 8: Example of various denoisers using diffusion sampling (except MMSE(Self Sup.) and MMSE(Sup.) columns) on NBU dataset with training and test noise level $\sigma_n = \sigma_t = 0.075$.

## A.3 EXTENSION TO LINEAR INVERSE PROBLEMS

We provide example reconstructions for the inpainting and demosaicing tasks in Figures 11 and 12 respectively.

## A.4 INFERENCE PROCEDURE

For diffusion sampling in our experiments we use a slightly modified version of the samplers proposed in Karras et al. (2022) by conducting sampling in measurement space. We show the inference procedure in Algorithm 1.

---

**Algorithm 1** Equivariant Sampling Inference

**Require:** $D_{\boldsymbol{\theta}}(\cdot, \sigma)$, $\{\sigma_K = \sigma_n, \ldots, \sigma_1 = \sigma_{\min}\}$, $\mathbf{y}$
1: $\mathbf{y}_{\text{next}} = \mathbf{y}$
2: **for** $i \in \{K, \ldots, 1\}$ **do**
3: $\quad \mathbf{y}_{\text{cur}} = \mathbf{y}_{\text{next}}$
4: $\quad \hat{\mathbf{x}} = \left(\mathbf{x}_{\text{cur}} - D_{\boldsymbol{\theta}}(\mathbf{A}^\top \mathbf{y}_{\text{cur}}, \sigma_i)\right) / \sigma_i$
5: $\quad \mathbf{y}_{\text{next}} = \mathbf{y}_{\text{cur}} + 2(\sigma_{i+1} - \sigma_i)\mathbf{A}\hat{\mathbf{x}}$
6: $\quad \boldsymbol{\eta} \sim \mathcal{N}(\mathbf{0}, \mathbf{I})$
7: $\quad \mathbf{y}_{\text{next}} = \mathbf{y}_{\text{next}} + \sqrt{2(\sigma_{i+1} - \sigma_i)\sigma_i}\, \boldsymbol{\eta}$
8: $\quad$ **if** $i > 1$ **then**
9: $\quad\quad \hat{\mathbf{x}}' = (\mathbf{x}_{\text{next}} - D_{\boldsymbol{\theta}}(\mathbf{A}^\top \mathbf{y}_{\text{next}}, \sigma_{i+1}))/\sigma_{i+1}$
10: $\quad\quad \mathbf{y}_{\text{next}} = \mathbf{y}_{\text{cur}} + \frac{1}{2}(\sigma_{i+1} - \sigma_i)\mathbf{A}(\hat{\mathbf{x}} + \hat{\mathbf{x}}')$
11: **return** $\hat{\mathbf{x}}$

---

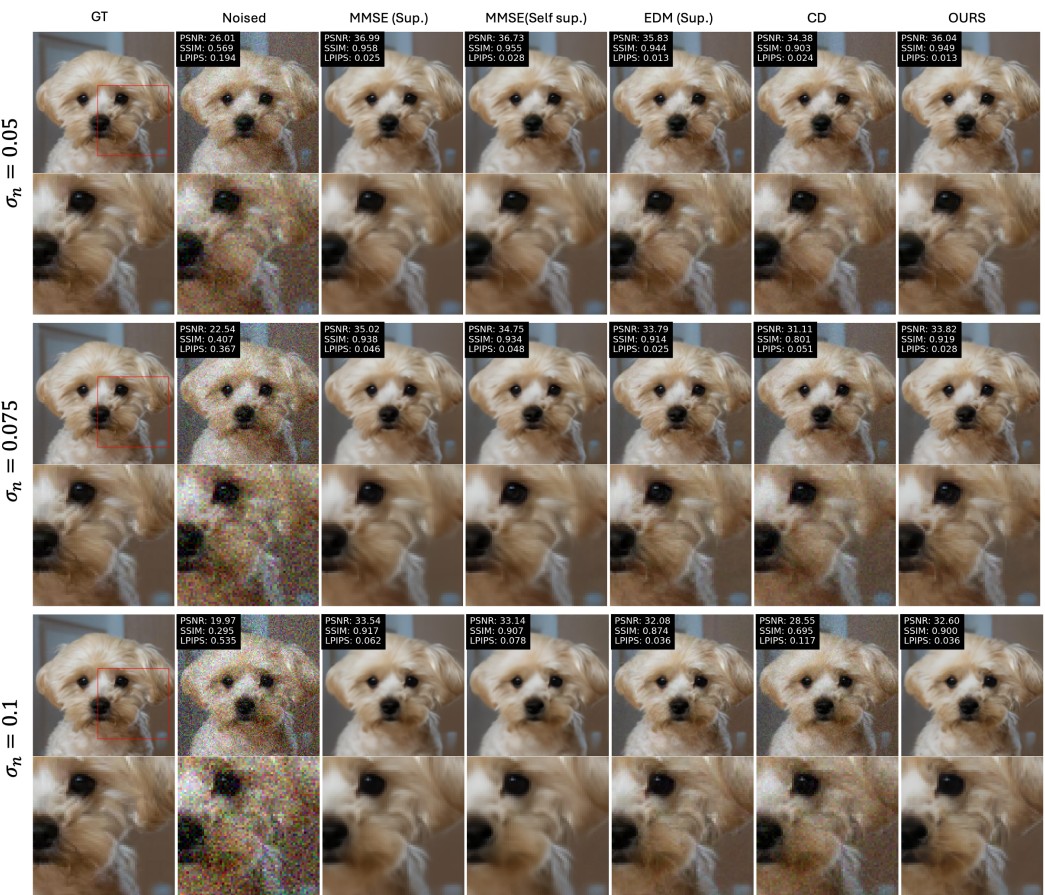

Figure 9: Example reconstructions for different training + inference noise levels on AFHQ.

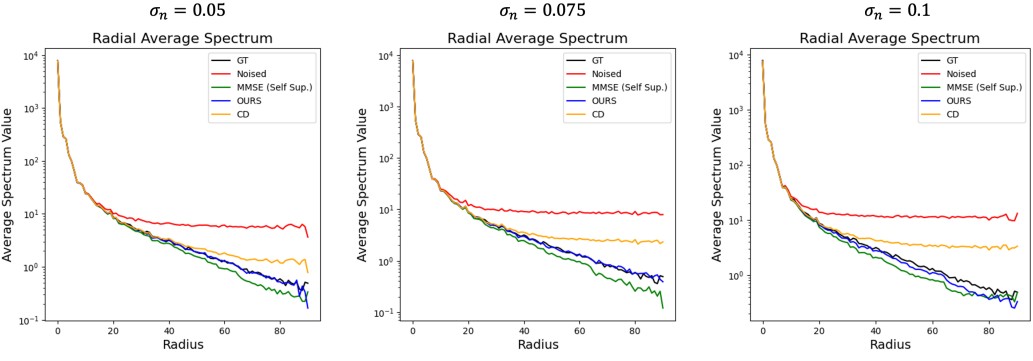

Figure 10: Radial Spectrum of images in Figure 9.

## A.5 PATCH NORMS

To investigate the assumption that many real image distributions are approximately scale-invariant we plot the histogram of patch-wise norms for several image distributions using various patch sizes in Figure 13. We see that, in fact, we have a spread in energy within each dataset which implies we may be observing a dataset that exhibits weak invariance.

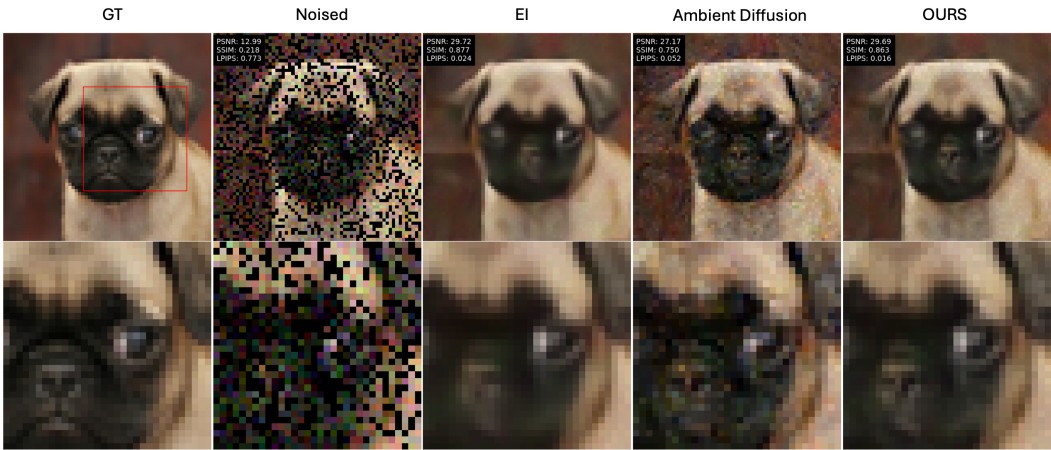

Figure 11: AFHQ inpainting example where the models are all trained in a self-supervised fashion.

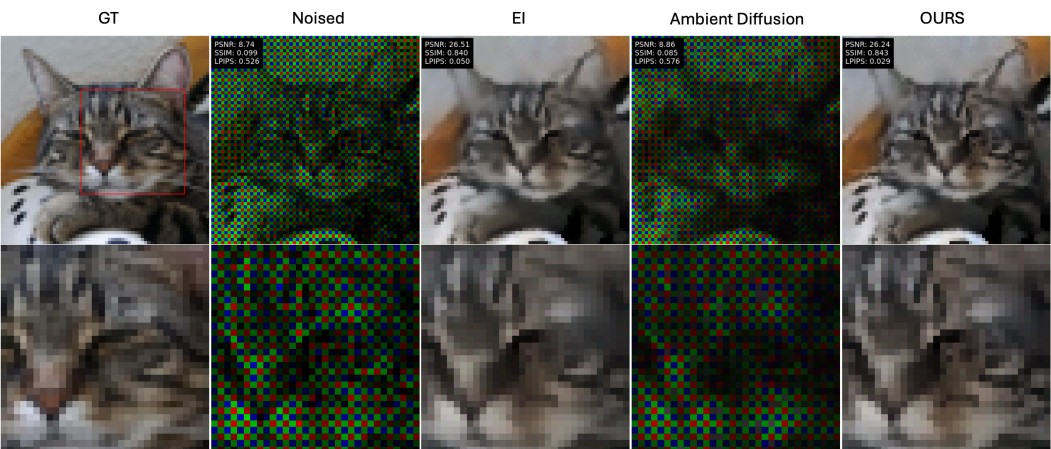

Figure 12: AFHQ demosaic example where the models are all trained in a self-supervised fashion.

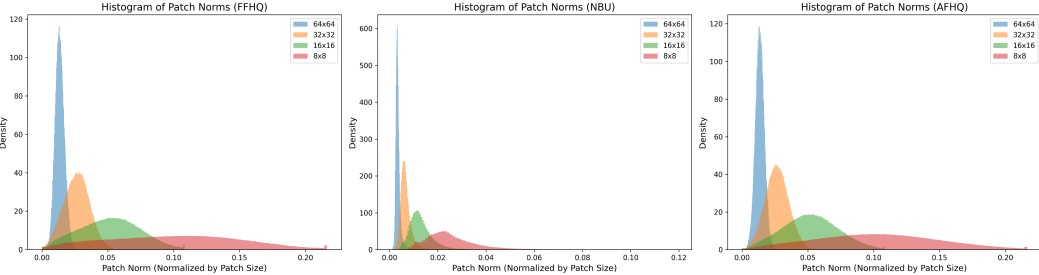

Figure 13: Histogram of image patches for each image distribution.

## A.6 BLIND DENOISING

Although our original design is for non-blind denoising, blind denoising is a potentially impactful extension for this work. To understand how our method degrades with improper specification of the dataset noise, we have included ablations for how our method preforms on the AFHQ denoising task with $\sigma_n = 0.075$ when we correctly ($\sigma_{exp} = 0.075$) and incorrectly

$(\sigma_{exp} = 0.01, 0.05, 0.08, 0.10, 0.15)$ estimate the noise level before training time (see fig. 14). Performance of self-supervised denoisers is known to be closely tied to correctly estimating the noise level of the training data (Tachella et al., 2025a) which we see in fig. 14.

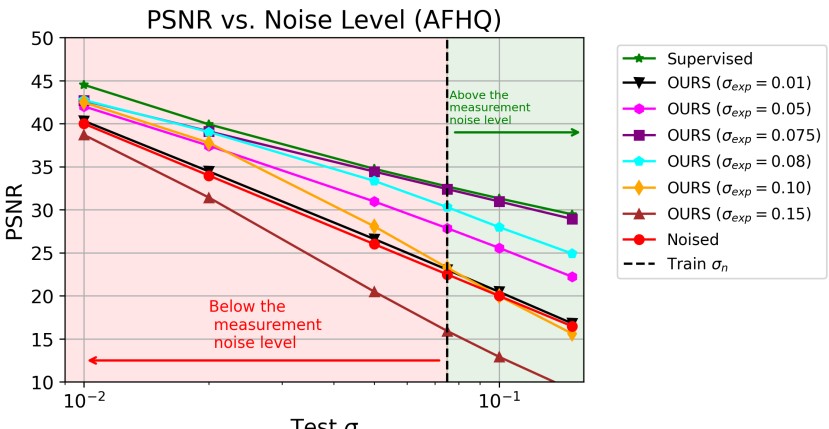

Figure 14: Denoising performance when the incorrect training noise level is used.

To solve this problem we have leveraged the UNSURE loss (Tachella et al., 2025a) in combination with our proposed loss giving the following training objective:

$$\min_{\theta} \max_{\sigma_n} \mathbb{E}_{\alpha,\mu} \left\{ \|\alpha \mathbf{y} + \mu \mathbf{1} - D_{\theta}(\alpha \mathbf{y} + \mu \mathbf{1}, \alpha \sigma_n)\|^2 + 2(\alpha \sigma_n)^2 \operatorname{div} D_{\theta}(\alpha \mathbf{y} + \mu \mathbf{1}, \alpha \sigma_n) \right\}, \quad (14)$$

We trained a denoiser on the same AFHQ $\sigma_n = 0.075$ task except we initialize our estimate for $\sigma_n = 0.01$ and treat it as a learnable parameter in (14). The resulting denoiser performance is shown in fig. 15. Here we see that although in the blind denoising case we do not completely bridge the gap with the non-blind version of our denoiser, we do improve considerably across noise levels.

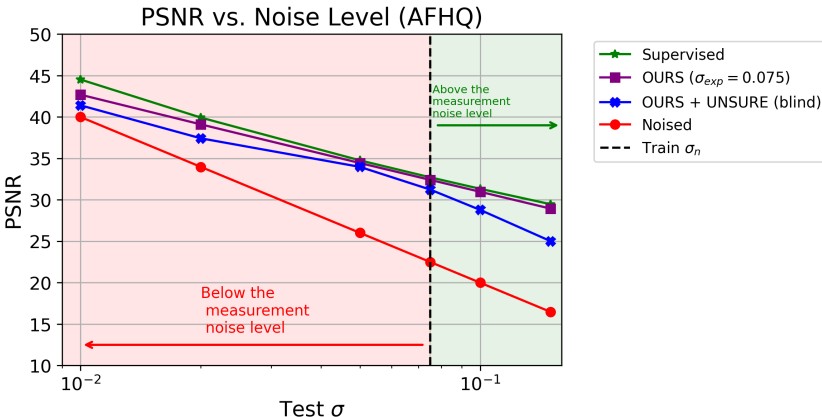

Figure 15: Denoising performance with UNSURE inspired loss in (14) compared to our method using the true $\sigma_n$.

## A.7 TRAINING SCHEMES

To investigate how sampling schemes, at training time, on $\alpha$ and $\mu$ affect the denoiser performance we compared four models: **(1)** original denoiser where $\alpha, \mu \sim U$ (uniform) **(2)** denoiser where $ln(\sigma_n \alpha) \sim \mathcal{N}(-1.2, 1.2^2)$ which was proposed for training diffusion models in (Karras et al., 2022) (edm) **(3)** denoiser where $\alpha \sim U$ and $\mu = 0$ is fixed at training time ($\mu = 0$) **(4)** a denoiser where $\alpha \sim U(0.5, 1)$ instead of $U(0, 1)$ (narrow). We show the results in fig. 16. Here we see the

sampling procedure for $\alpha$ does not seem to affect the performance of the model as long as we cover the full range of noise levels we wish to test on. We also see that not using $\mu$ leads to degraded performance as the noise level decreases.

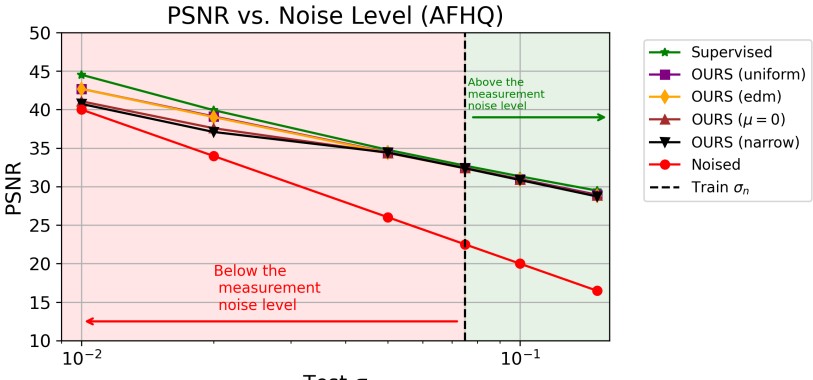

Figure 16: Denoising performance between models trained using different sampling procedures for $\alpha$ and $\mu$.

## A.8 INFERENCE NOISE SCHEDULING

Here we provide a look at how $\sigma_t$ schedules and the number of inference steps (K) affects the performance of our denoiser. We test our method with $K = 5, 25, 100$ steps using three different schedules for $\sigma_t$. Specifically, we use alternative schedules for $\sigma_t$ by varying $\gamma$ in (12). We tested performance at each step count with $\gamma = 0.5, 1, 7$. See fig. 17 examples of the noise schedules at $K = 5, 25, 100$. Figure 18 shows plots of performance with varying step counts $K$ and noise schedules $\sigma_t$ using our trained denoiser on the AFHQ dataset with $\sigma_n = 0.075$. We found that $\gamma = 7$ provided the best performance in the fewest number of steps and only minimal improvements are obtained with more than 25 inference steps using $\gamma = 7$. Our findings are consistent with prior work (Karras et al., 2022) which we based our inference noise schedule on.

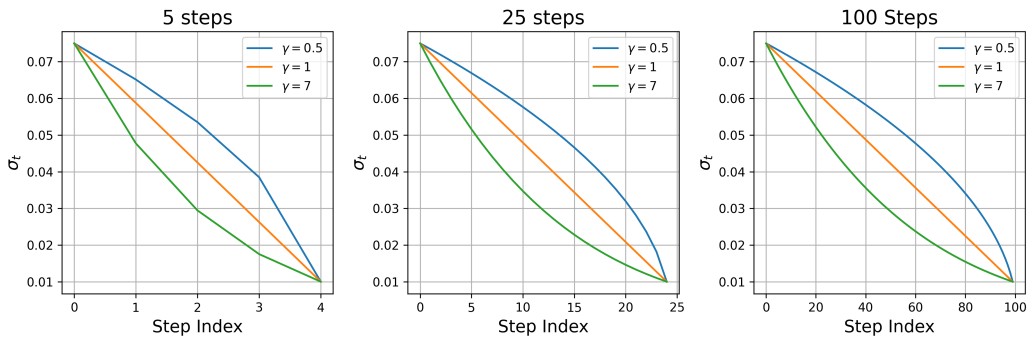

Figure 17: Example noise schedules for $\gamma = 0.5, 1, 7$.

## A.9 MRI RECONSTRUCTION

To show that our method can generalize to other linear inverse problems we trained a model for MRI reconstruction. Here the data is complex valued and under-sampled using a 2D fourier transform followed by a single under-sampling mask and additive gaussian measurement noise ($\sigma_n = 0.1$). We use single coil brain data from the fastMRI dataset (Zbontar et al., 2019). Our training dataset consisted of $20,000$ noisy measurements. We chose rotations to use as the group of transformation to enforce equivariance for both our method and EI. Numerical results across a test set of 700 samples

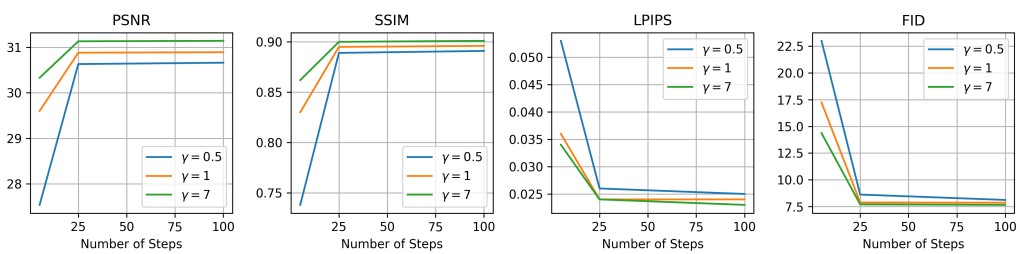

Figure 18: Sampling metrics using various $\gamma = 0.5, 1, 7$ and step counts $K = 5, 25, 100$ in (12).

Table 3: Fourier under-sampling on single coil fastMRI $64 \times 64$.

| Task | $\sigma_n$ | Solver | Sampler | Self Sup. | PSNR ($\uparrow$) | SSIM ($\uparrow$) | LPIPS ($\downarrow$) | FID ($\downarrow$) |
|------|-----------|--------|---------|-----------|-------------------|-------------------|---------------------|--------------------|
| MRI Recon. | 0.075 | EI | | $\checkmark$ | **23.16** | 0.796 | 0.043 | 86.94 |
| | | OURS | $\checkmark$ | $\checkmark$ | 22.22 | **0.812** | **0.027** | **72.88** |

using our method compared to EI are shown in table 3. An example reconstruction is presented in fig. 19 where we see better perceptual quality in the image contrast when comparing our technique to EI.

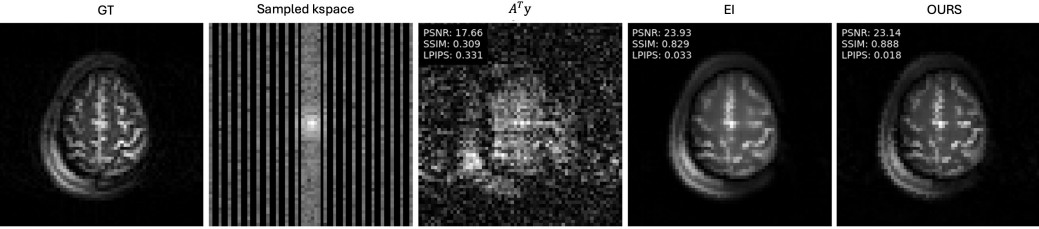

Figure 19: MRI reconstruction example.

## A.10 EQUIVARIANCE THROUGH ARCHITECTURE

To investigate the performance of equivariant architectures in self-supervised settings we trained a normalization-equivariant architecture (Herbreteau et al., 2024) with SURE. We compared this method on the same AFHQ $\sigma_n = 0.075$ dataset used previously. Figure 20 shows how our denoiser, and those in the main paper, compare to using an equivariant denoising architecture. We see that although the equivariant architecture does help marginally at lower noise levels the performance drop is much greater than our approach. If we look at the example denoising images in fig. 21 we see that the equivariant architecture heavily over smooths the lower noise corrupted images compared to our method.

## A.11 PROSPECTIVE LOW-FIELD MRI DENOISING

To further demonstrate that our method works on real sensor data we applied our method to an in-house MRI dataset of 62 subjects whose wrists were scanned on a 1-Tesla permanent magnet MRI scanner with Institutional Review Board approval and informed consent. The scans were fully sampled using a multi-echo spin-echo (MESE) sequence, and thus the inherent SNR in each sample variable, with measured noise levels $\sigma_n \in [0.04, 0.06]$. Our training set consisted of $30,000$ noisy images (see fig. 22). The noise level was estimated for each image in the training dataset by sampling background pixels (top left or bottom right patch) and calculating the standard deviation. Our denoiser was trained directly on the inverse fourier transform of the raw measurements. A self-supervised denoising example using our method is shown in fig. 23. Since all the measurements are corrupted with noise in this real dataset we cannot provide image quality metrics. However, we

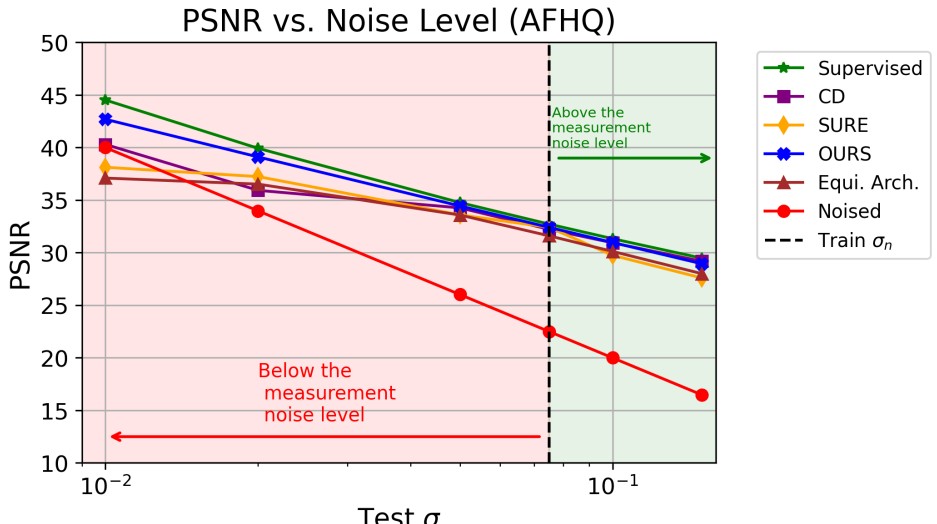

Figure 20: Denoiser comparison on AFHQ $\sigma_n = 0.075$ denoising task.

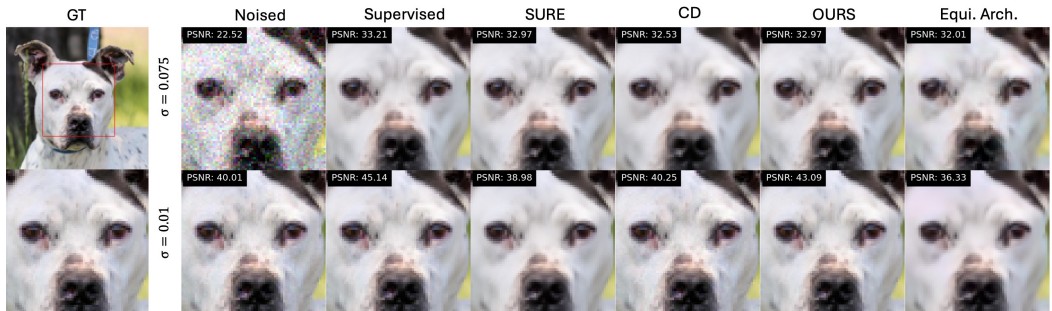

Figure 21: Example Denoiser comparison on AFHQ denoising task for inference at the training noise level $\sigma = 0.075$ and below the training noise level $\sigma = 0.01$.

visually observe that our method is removing measurement noise and can provide both a plausible sample along with a variance map calculated from sampling 10 different reconstructions.

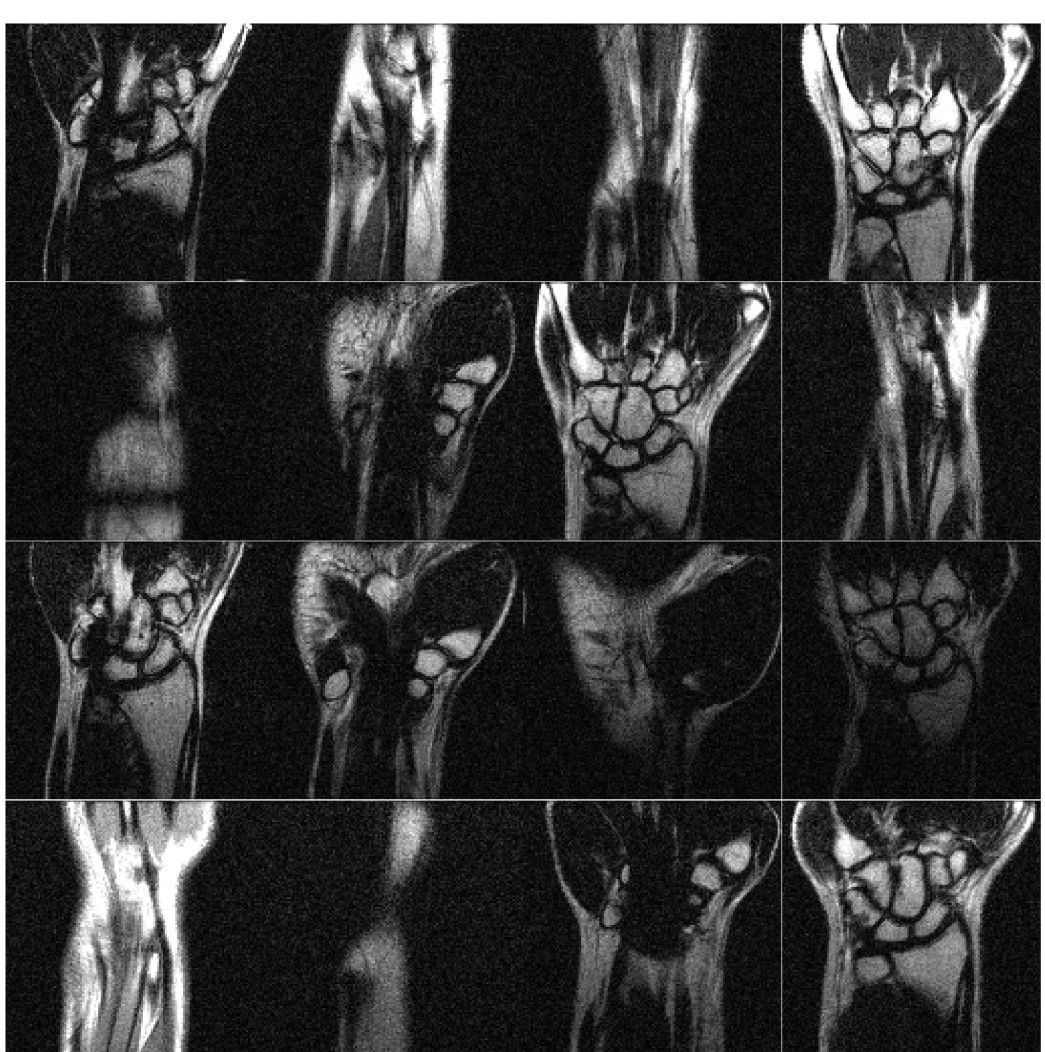

Figure 22: Prospectively collected MRI training data for self-supervised denoising experiments.

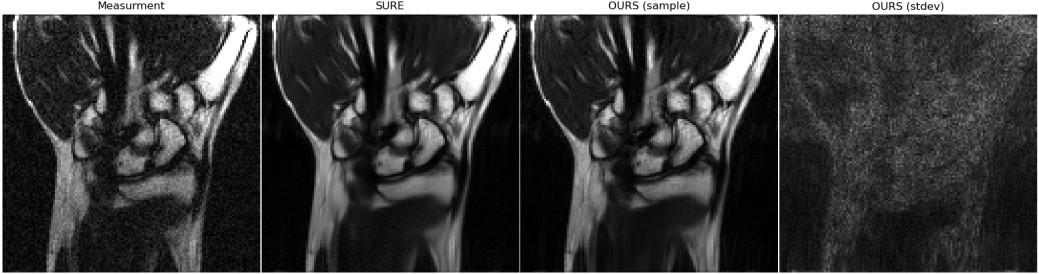

Figure 23: Prospectively collected MRI test data denoised using our self-supervised trained denoiser + sampler.