# OpenReview forum: "Normalization-equivariant Diffusion Models: Learning Posterior Samplers From Noisy And Partial Measurements"
_ICLR.cc/2026/Conference — Submitted to ICLR 2026_

### Official Review · Reviewer_wSF6 · 2025-10-31

**Soundness:** 4
**Presentation:** 3
**Contribution:** 3
**Rating:** 4
**Confidence:** 4

**Summary:**

This paper tackles the challenging problem of training diffusion models from a single set of noisy and incomplete measurements. The authors introduce a normalization equivariance property to modify the SURE loss, enabling a denoiser trained only at noise level $\sigma_n$ to generalize to levels $\sigma < \sigma_n$. This key insight, when combined with the Equivariant Imaging framework, allows the model to learn a posterior sampler from a single, fixed, and rank-deficient measurement operator. The method is validated on denoising, inpainting, and demosaicing, showing strong perceptual quality against self-supervised baselines.

**Strengths:**

1. The paper tackles an important problem of training generative models from single-operator, noisy, and incomplete measurements.
2. The identification and exploitation of "normalization equivariance" as a mechanism to bridge the gap and train a denoiser for $\sigma < \sigma_n$ using only data at $\sigma_n$ is a clever and original contribution.
3. The empirical results are comprehensive and promising.

**Weaknesses:**

1. The authors propose a loss-based enforcement and state, "we find that this leads to better performance for self-supervised learning". This is a strong claim, but it is not substantiated with an experimental comparison. A key missing ablation would be to train the architecturally-equivariant network from Herbreteau et al. (2024) using the SURE loss and compare it against the proposed $\mathcal{L}_{NE,SURE}$.
2. The proposed loss $\mathcal{L}_{NE,SURE}$ samples $\alpha \sim \mathcal{U}(0,1)$ and $\mu \sim \mathcal{U}(0,1)$. This choice seems arbitrary. How sensitive is the method to this choice? Would a different distribution (e.g., log-uniform for $\alpha$) be more effective? Some justification or ablation on this design choice is needed.
3. In Figure 4, the authors claim the MSE scales approximately as $\sigma_t^2/\sqrt{N}$. However, the $C/\sqrt{N}$ line does not appear parallel to several of the empirical lines. This claim seems to be a slight over-simplification of the empirical result shown.

**Questions:**

1. What is the rationale for sampling $\alpha, \mu \sim \mathcal{U}(0,1)$ in Eq. 8? Have the authors explored the sensitivity of the method to this sampling distribution?
2. Is there a more competitive generative self-supervised baseline for the single-operator setting apart from Ambient Diffusion?
3. Learning from corrupted data is very timely. I would recommend that the authors also discuss their method in the context of other recent works tackling similar problems, which may offer complementary insights:
    - Restoration Score Distillation: From Corrupted Diffusion Pretraining to One-Step High-Quality Generation, arxiv, 2025
    - An Expectation-Maximization Algorithm for Training Clean Diffusion Models from Corrupted Observations, NeurIPS 2024
    - Learning Diffusion Priors from Observations by Expectation Maximization, NeurIPS 2024

---

> ### Author Response · Authors · 2025-11-20
>
> We thank the reviewer for taking the time to leave valuable feedback and comments on our paper.
>
> We compared our approach with the normalization equivariant architecture from Herbreteau et al. (2024) trained using a SURE loss in section A.10 in the appendix. We see that although the equivariant architecture does help marginally at lower noise levels the performance drop is much greater than our approach. In the denoising example provided we see that the equivariant architecture heavily over smooths the lower noise corrupted images compared to our method.
>
> We have added an investigation into the effects of different sampling schemes for $\alpha$ and $\mu$ at training time (see section A.7 in the appendix). We showed that (1) using $\mu\neq 0$ greatly helps our method at $\sigma<\sigma_n$ (2) Our method is not very sensitive to the sampling distribution for $\alpha$ as long as $\alpha$ is sampled from the full range $\alpha\in(0,1)$.
>
> The reviewer was correct to point this out oversimplification in the figure 4 plot. We have added a line for $C/N^{1/3}$ for comparison to the main paper figure.
>
> These papers are highly relevant to the space of self-supervised learning generative models from corrupted data, but they consider a substantially different setup. The listed methods assume the availability of multiple measurement operators that jointly cover the full range space, so that the data provides all the information required for training, whereas our approach does not. We attempt to solve the case where measurements come from a single measurement process, reflecting the data generated by a specific sensing instrument implementing a specific measurement protocol. This leads to measurements that are potentially very incomplete from an information viewpoint, a difficulty that we address by exploiting symmetries and invariance properties in the problem. To illustrate this point, we compared our approach to Ambient Diffusion as a representative of the state-of-the-art for this class of multi-operator learning approaches. Crucially, these methods also differ significant from our approach in the manner in which we scale to below the natural noise level. Our reported experiments show that the our approach is far more effective for the setting where all data are measured from a single instance of the measurement operator.

---

### Official Review · Reviewer_uUHJ · 2025-10-31

**Soundness:** 3
**Presentation:** 3
**Contribution:** 2
**Rating:** 4
**Confidence:** 3

**Summary:**

This paper proposes a diffusion-based image restoration framework that learns entirely from noisy measurement data obtained through a single operator. By embedding the normalization-equivariance property into the standard SURE loss, the method enables training without clean ground truth and achieves competitive performance across denoising, demosaicing, and inpainting, along with comparisons with the state of the art.

**Strengths:**

- Good motivation and no ground-truth requirement. The paper presents a theoretical framework to train diffusion-based denoisers and posterior samplers without any access to clean data, offering a practical and scalable solution for real-world denoising and inverse problems where ground-truth supervision is infeasible.

- Novel theoretical insight and finding (Normalization-Equivariance). The authors contribute by formalizing a new normalization-equivariant property of MMSE denoisers, providing both analytical justification and architectural grounding for learning across unseen noise levels.

**Weaknesses:**

- Theoretical assumptions are idealized. The normalization-equivariance assumption relies on the prior being approximately positively homogeneous and factorable into radial and angular components. This theoretical assumptions may not be practical in real world setting.

- Restricted to gaussian noise and linear operators. The method is formulated specifically for additive Gaussian noise and linear forward operators. Its applicability to non-Gaussian, structured, or data-dependent degradations (e.g., Poisson, compression artifacts, MRI nonlinearity) remains unclear.

- Limited real-world dataset validation and ablation.
All experiments are conducted on synthetic corruption processes (Gaussian noise, masking, demosaicing) using curated datasets like FFHQ, AFHQ, and NBU. The method has not been evaluated on real-world restoration benchmarks (e.g., RainDrop, AllWeather, SIDD, BSD-Denoise), where uncontrolled degradations and sensor characteristics could significantly affect performance. More ablation and cross-domain testing would strengthen claims of generalization.

**Questions:**

Could the authors evaluate how the proposed normalization-equivariant diffusion framework performs on real-world degradation datasets (e.g., AllWeather, RainDrop, SIDD) where noise characteristics deviate from the assumed Gaussian model and amplitude–structure independence does not strictly hold?

---

> ### Author Response · Authors · 2025-11-20
>
> We thank the reviewer for their time spent providing feedback on our paper.
>
> We fully agree that assuming that the image prior factorizes fully as a radial and angular component - i.e., $p(x) = p(\frac{x}{\\\|x\\\||})p(\\\|x\\\|)$ - is a strong assumption. However, to motivate why this assumption holds approximately in practice, one can consider the decomposition $p(x) = p(\frac{x}{\\\|x\\\|} | \\\|x\\\|)p(\\\|x\\\|)$ which is always valid, and then contrast $p(\frac{x}{\\\|x\\\|} | \\\|x\\\|)$ with $p(\frac{x}{\\\|x\\\|})$ by considering what how much additional information about $x$ knowing the value of $\\\|x\\\|$ provides, when $x$ is an unknown high-dimensional image and $\\\|x\\\|$ is just it's norm. In words, the fact that recovering estimates of $x$ from $\\\|x\\\|$ is extremely challenging indicates that $p(x) \approx p(\frac{x}{\\\|x\\\|})p(\\\|x\\\|)$.
>
> We also agree that scale invariance for the image prior is a strong assumption. However, approximate scale invariance for the minimum mean squared error denoisers is a much milder assumption, notably because we we only require the approximation to hold in the range of scales of values of $\\\|x\\\|$ where the posterior distribution has most of its probability mass. In most cases of interest, the measurement $y$ has enough information to pin down the value of $\\\|x\\\|$, hence the range of values where the approximation should hold is quite narrow in practice. These important points will be clarified in the revised manuscript.
>
> In addition, our experiments show that many different natural/scientific image distributions we experimented upon (AFHQ, FFHQ, NBU, MRI) exhibit weak forms of invariance inherently, providing additional backing to this assumption and to why our method is able to learn accurate denoisers below the data noise level. We emphasize that we did not pre-process the image distributions used in this paper to purposefully make them invariant before noising the measurements. Our experiments on these datasets show that we are able to achieve SOTA performance by leveraging the weak invariance inherent to natural image distributions for denoising.
>
> As pointed out, our method is tailored to additive gaussian noise with a linear forward operator. This, however, is an important setting that is highly relevant to many computational imaging applications including MRI, radar imaging, ultrasound etc. Moreover, the linear additive Gaussian model provides a foundational step toward broader and more flexible models such as the exponential family approaches. We have added an experiment on single coil under-sampled + noised MRI reconstruction to the Appendix section A.9 to further demonstrate the generalization of our method. While our method cannot be immediately applied to non-linear/blind imaging tasks, our work sets the stage for future work to extend our technique to those scenarios.

---

> > ### Comment · Reviewer_uUHJ · 2025-11-25
> >
> > Thank you for the thorough and thoughtful author feedback. While I really appreciate author's insights and the strengthened discussion, they do not materially alter my overall assessment of the paper. I therefore maintain my original score. I will also revisit the discussion later to consider any further value points raised by other reviewers.

---

> > > ### Author Response · Authors · 2025-11-26
> > >
> > > We sincerely thank you for your reply. We are disappointed to hear that our response does not materially alter your assessment of the paper and that you will not reconsider your initial score. With due respect, we kindly request that you provide us with actionable feedback that you believe would merit a change in your review.
> > >
> > > In response to your point about the idealized nature of our assumptions, we pointed out that for all our experiments we did not artificially create datasets which fit the invariance assumption; rather, we showed our method leveraged the weak invariances inherently present in the datasets. This is both a novel and practical finding.
> > >
> > > We emphasize that our experimental setup for unsupervised image denoising exactly follows what previous SOTA methods do for their method training/validation (Metzler et al., 2020; Soltanayev & Chun, 2021; Aali et al., 2023, Daras et al., 2024b;a). Specifically, they leverage clean datasets and apply gaussian corruptions before training. We would like to point out that we have also applied our method to a wider variety of image datasets than many previous SOTA methods (AFHQ, NBU, FFHQ, MRI).
> > >
> > > We appreciate the suggestion to use specific real datasets, however they don't fit our setting and that of previous SOTA works in our space. Specifically:
> > >
> > > RainDrop: rain drop removal dataset which does not fit the linear inverse problem setting.
> > >
> > > AllWeather: adverse weather correction which does not fit the linear inverse problem setting.
> > >
> > > SIDD: image denoising where the noise is signal dependent and variable across acquisitions which does not fit our assumption of data from a single measurement system with a fixed noise level. We emphasize this is the assumption of previous SOTA work in our area as well.
> > >
> > > As for low-rank + noisy measurement systems, we demonstrated the general nature of our method on inpainting, demosaicing, and in response to your suggestion to bolster generalization claims, subsampled MRI.
> > >
> > > We want to emphasize that in the space of image restoration, linear inverse problems with gaussian noise are extremely important across many different areas (MRI, inpainting, demosaicing, super resolution, deblurring, CT, etc.) and popular SOTA methods are typically shown experimentally for such scenarios (Bahjat Kawar, et al. 2022,  Yuanzhi Zhu, et al. 2023).
> > >
> > > While we fully acknowledge that many of the real datasets listed do not fit the setting for our method (due to unknown forward process or different noise statistics) we wish to point out that our method does fit for the real-world inverse problems that we showed across the 4 different datasets we tested. These are relevant problems that are of interest to the community as demonstrated by the extensive collection of previous works on the subject of linear inverse problems + additive gaussian noise. We acknowledge the significance of the applications you identified; however, as with many techniques, our initial emphasis is on the broad and impactful domain of linear inverse problems, which lays the groundwork for addressing more specific problems afterward.
> > >
> > > Thank you again for taking the time to review our paper and provide constructive feedback to improve its quality.

---

> > > > ### Author Response · Authors · 2025-11-28
> > > >
> > > > To further demonstrate that our method works on real sensor data we applied our method to an in-house MRI dataset of 62 subjects whose wrists were scanned on a 1-Tesla permanent magnet MRI scanner with Institutional Review Board approval and informed consent. The scans were fully sampled using a multi-echo spin-echo (MESE) sequence, and thus the inherent SNR in each sample variable, with measured noise levels between [0.04, 0.06]. A self-supervised denoising example using our method is shown in appendix section A.11. Since all the measurements are corrupted with noise in this real dataset we cannot provide image quality metrics. However, we  visually observe that our method is removing measurement noise and can provide both a plausible sample along with a variance map calculated from sampling 10 different reconstructions.

---

### Official Review · Reviewer_ZjQq · 2025-11-01

**Soundness:** 3
**Presentation:** 2
**Contribution:** 2
**Rating:** 6
**Confidence:** 3

**Summary:**

The paper introduces a novel approach for training diffusion models (DMs) for image restoration using only noisy measurement data from a single operator. The authors show that DMs and MMSE denoisers exhibit a weak form of scale equivariance, linking signal rescaling to changes in noise intensity. Building on this insight, they develop a denoising score-matching strategy that generalizes to lower noise levels than those seen during training. To handle incomplete and noisy data, their method is integrated with equivariant imaging, and experiments on denoising, demosaicing, and inpainting demonstrate its effectiveness compared to state-of-the-art approaches.

**Strengths:**

1. The method tackles a challenging problem of image restoration using only corrupted data which can be useful in some scenarios.

2. The method achieves better results compared to the existing methods.

**Weaknesses:**

1. I think the method can only be applied to some specific degradations like noise and masks.

**Questions:**

1. Can the method be extended in the case of data corrupted with multiple degradations?

2. Can the method be applied to more challenging degradations like blur for example?

---

> ### Author Response · Authors · 2025-11-20
>
> We thank the reviewer for spending time to provide valuable comments on our manuscript.
>
> Our method can be applied to imaging scenarios where the forward operator is linear and known at training/inference time. This fits many other imaging scenarios than the ones we showed in our main paper submission such as MRI. To further demonstrate this, we have included a new result showing our method for MRI reconstruction (see section A.9 in appendix) which is Fourier under sampling. The method can be extended to data with multiple degradations if they can be represented as a linear operator. We have also extended our method in the appendix section A.6 to handle the blind denoising case where we don't know the exact noise level at training time.

---

### Official Review · Reviewer_WyJL · 2025-11-04

**Soundness:** 2
**Presentation:** 3
**Contribution:** 2
**Rating:** 4
**Confidence:** 3

**Summary:**

The paper proposes **Normalization-Equivariant Diffusion Models** for learning posterior samplers using only noisy and/or partial measurements from a single operator. The key idea is to exploit a *scale (normalization) equivariance* property of MMSE denoisers to transfer supervision from the measurement noise level to lower noise levels, enabling conditional score learning without clean data. Concretely, the authors embed this property into a modified SURE objective (NE-SURE) and then plug the resulting denoiser into a diffusion sampler. They further combine this with Equivariant Imaging to handle rank-deficient single-operator inverse problems (e.g., inpainting, demosaicing). Experiments on FFHQ/AFHQ/NBU show improved perceptual metrics over self-supervised baselines and competitive results to supervised counterparts.

**Strengths:**

Shows that a weak form of scale equivariance links amplitude rescaling and noise level, and operationalizes it via NE-SURE (Eq. 8) to learn below the measurement noise—critical for diffusion posterior sampling without clean data. A theorem formalizes when MMSE denoisers are approximately normalization-equivariant under a factorized prior, lending plausibility to the rescaling argument. On denoising, the method maintains strong performance below the training noise and improves perceptual metrics over self-supervised baselines; on inverse problems, it outperforms EI/Ambient Diffusion in FID/LPIPS while EI remains better in distortion metrics—consistent with perception–distortion trade-offs.

**Weaknesses:**

1. The approximate scale-equivariance relies on structural assumptions about the image prior (factorization into norm and direction); discussion is persuasive but still idealized, and empirical ablations that probe violation of these assumptions (e.g., strongly non-homogeneous priors) are limited.
2. Training and evaluations assume additive Gaussian noise with known level; robustness to modest model mismatch (e.g., mis-specified σ, signal-dependent noise) is not studied.
3. While the EI fusion is elegant, demos focus on synthetic masks/Bayer patterns at one noise level; broader operators (e.g., blur, Fourier subsampling) or real sensor pipelines would strengthen claims of generality.

**Questions:**

1. How sensitive is NE-SURE to under/over-estimating the measurement noise? Please add curves where training assumes ωₙ′ ≠ true ωₙ.
2. Beyond (ε, μ) sampling in Eq. (8), what happens if μ=0 or if ε sampling range narrows? Does performance degrade smoothly?
3. Can you show results for additional single operators (e.g., spatial blur, partial Fourier) to complement inpainting/demosaicing?
4. How do FID/LPIPS trade with number of reverse-SDE steps K and ω-schedules for the same trained denoiser?

---

> ### Author Response · Authors · 2025-11-20
>
> We thank the reviewer for their time and providing valuable feedback.
>
> An important point is made about the application of our method to blind denoising scenarios where the noise level is not exactly known a training time. We have run experiments (see Section A.6 in appendix) which explore how our method performs when the noise level is mis-specified at training time. We see that mis-specifications in the noise level can indeed adversely affect our method. To remedy this, we proposed a modification to our training loss leveraging previous work UNSURE which trained denoisers in the blind setting at a single noise level using a min-max objective. We adapted this loss to ours so we could learn at and below the unknown natural noise level in the blind setting. Using this loss we trained a new denoiser which greatly increased the performance of our method in the blind setting.
>
> We have added an investigation into the effects of different sampling schemes for $\alpha$ and $\mu$ at training time (see section A.7 in the appendix). We showed that (1) using $\mu\neq 0$ greatly helps our method at $\sigma<\sigma_n$ (2) Our method is not very sensitive to the sampling distribution for $\alpha$ as long as $\alpha$ is sampled from the full range $\alpha\in(0,1)$.
>
> We have added additional experiments investigating the effects of noise schedules at inference time (see section A.8 in Appendix). There we show that using the noise schedule from our main results preformed better in fewer steps than other noise schedules and only requires $\sim25$ steps to reach near maximal performance.
>
> To show that our method can be generalized to other linear inverse problem settings we trained a new model for MRI reconstruction. Here we under-sampled a dataset of single-coil MRI data with a single sampling mask + additive noise and trained our method and EI for comparison. We observed that, as with the other imaging experiments, our method outperforms EI in perceptual quality and even in some distortion metrics like SSIM for the sub-sampled MRI case. These experiments have been added to the appendix in section A.9.
>
> We fully agree that scale invariance for the image prior is a strong assumption. However, approximate scale invariance for the minimum mean squared error denoisers is a much milder assumption, notably because we we only require the approximation to hold in the range of scales of values of $||x||$ where the posterior distribution has most of its probability mass. In most cases of interest, the measurement $y$ has enough information to pin down the value of $||x||$, hence the range of values where the approximation should hold is quite narrow in practice. This important point will be clarified in the revised manuscript.
>
> In addition, our experiments show that many different natural/scientific image distributions we experimented upon (AFHQ, FFHQ, NBU, MRI) exhibit weak forms of invariance inherently, providing additional backing to this assumption and to why our method is able to learn accurate denoisers below the data noise level. We emphasize that we did not pre-process the image distributions used in this paper to purposefully make them invariant before noising the measurements. Our experiments on these datasets show that we are able to achieve SOTA performance by leveraging the weak invariance inherent to natural image distributions for denoising.

---

### Meta-Review · Area_Chair_Pvw4 · 2025-12-19

**Summary:**

The paper proposes a method for training diffusion models using only noisy measurements from a single operator by exploiting a property termed normalization equivariance. By integrating this property into a modified SURE objective, the authors aim to enable learning at noise levels lower than the training data. While the reviewers acknowledged the novelty of the theoretical insight and the clever integration with equivariant imaging, the consensus is that the method relies on strong, idealized assumptions.. Furthermore, concerns were raised regarding the method's limitation to additive Gaussian noise and linear operators, with reviewers finding the validation on synthetically corrupted data insufficient to demonstrate robustness in real-world scenarios. The authors provided a rebuttal that addressed several empirical shortcomings, including new experiments on blind denoising and MRI reconstruction, which clarified technical queries regarding parameter sensitivity. However, despite these substantial improvements, the rebuttal did not sufficiently outweigh the fundamental reservations about the method’s theoretical generalizability to complex, non-idealized imaging conditions.

**Reviewer Concerns:**

Addressed in the rebuttal
- The authors successfully addressed concerns regarding the choice of sampling distributions for the NE loss parameters ($\alpha, \mu$) by providing ablations showing the method is robust to these choices.
- The authors provided a comparison against architecturally equivariant networks, showing their method's superior performance.
- The authors extended the method to handle unknown noise levels during training, addressing a key practical concern raised by one reviewer.
- The authors corrected the theoretical scaling claim in Figure 4.

Outstanding or partially addressed in the rebuttal:
- The core assumption of normalization equivariance relies on the image prior being factorizable (independence of norm and direction), which is an idealization that may not hold for complex natural image distributions.
- Despite new experiments on MRI data, the method was not sufficiently validated on standard real-world benchmarks (e.g., SIDD, RainDrop) where noise is signal-dependent or non-Gaussian. The restriction to linear forward operators remains a significant limitation for general image restoration tasks.

**Reviewer Scores:**

- Reviewer WyJL (Initial Score: 4): Likely would remain a 4 (or move to a 5). While the authors addressed the blind denoising question, the reviewer's fundamental concern about the idealized structural assumptions of the prior was not really resolved by the rebuttal arguments.
- Revieer ZjQq (Initial Score: 6): Likely would remain a 6.
- Reviewer uUHJ (Iniital Score: 4): Likely would remain a 4. This reviewer explicitly stated after the rebuttal that the author's feedback didn't materially alter their assessment and they maintained their original score.
- Reviewer wSF6 (Score: 4): Likely would increase to a 5 or 6. The authors addressed this reviewer's specific requests for architectural baselines and corrected the figure error. This reviwer had concerns that were more empirical than theoretical.

---

### Decision · Program_Chairs · 2026-01-26

Reject